


# Nitrate chemistry in the northeast US part I: nitrogen isotope seasonality tracks nitrate formation chemistry

Claire Bekker[1,a†], Wendell W. Walters[2*†], Lee T. Murray[3], Meredith G. Hastings[1,2]

[1]Department of Earth, Environmental, and Planetary Sciences, Brown University; Providence, RI 02912, USA

[2]Institute at Brown for Environment and Society, Brown University; Providence, RI 02912, USA

[3]Department of Earth and Environmental Sciences, University of Rochester; Rochester, NY 14627, USA

[a]Now at Department of Environmental Health Sciences, University of California Los Angeles; Los Angeles, CA 90095, USA

[†]These authors contributed equally to this work.

*Correspondence to*: Wendell W. Walters (wendell_walters@brown.edu)

**Abstract.** Despite significant precursor emission reductions in the US over recent decades, atmospheric nitrate deposition remains an important terrestrial stressor. Here we utilized statistical air mass back trajectory analysis and nitrogen stable isotope deltas ($\delta(^{15}N)$) to investigate atmospheric nitrate spatiotemporal trends in the northeastern US from samples collected at three US EPA Clean Air Status and Trends Network (CASTNET) sites from December 2016-2018. For the considered sites, similar seasonal patterns in nitric acid ($HNO_3$) and particulate nitrate ($pNO_3$) concentrations were observed with spatial differences attributed to nitrogen oxide ($NO_x$) emission densities in source contributing regions that were typically ≤1000 km. Significant spatiotemporal $\delta(^{15}N)$ variabilities in $HNO_3$ and $pNO_3$ were observed with higher values during winter relative to summer, like previous reports from CASTNET samples collected in the early 2000s for our study region. In the early 2000s, $\delta(^{15}N)$ of atmospheric nitrate in the Northeast US had been suggested to be driven by $NO_x$ emissions; however, we did not find significant spatiotemporal changes in the modeled $NO_x$ emissions by sector and fuel type or $\delta(^{15}N, NO_x)$ for the source regions of the CASTNET sites. Instead, the spatiotemporal trends were driven by $\delta(^{15}N)$ fractionation associated with nitrate



formation. Under the field conditions of low $NO_x$ relative to $O_3$ concentrations and when $\delta(^{15}N, NO_x)$ emission sources do not have significant variability, we demonstrate that $\delta(^{15}N)$ of atmospheric nitrate can be a robust tracer for diagnosing nitrate formation.

# 1 Introduction

Nitrogen oxides ($NO_x$ = NO + $NO_2$) are a significant source of air pollution derived from electricity generation, industrial processes, vehicle emissions, biomass burning, lightning, and microbial activity in soils (Jaeglé et al., 2018, 2005; Delmas et al., 1997). $NO_x$ emissions have an important impact on climate and human and ecosystem health due to their influence on atmospheric oxidation chemistry and production of atmospheric nitrate ($tNO_3$ = nitric acid ($HNO_3$) + particulate nitrate ($pNO_3$)) (Galloway et al., 2004; Zhang et al., 2003; Frost et al., 2006; Pinder et al., 2012). $NO_x$ chemistry facilitates the production of atmospheric oxidants, including ozone ($O_3$) and hydrogen oxide radicals ($HO_x$ = OH + $HO_2$), which defines the tropospheric oxidation capacity (Bloss et al., 2005; Prinn, 2003). These oxidants play an important role in the removal of trace gases and formation of particulate matter, with important consequences for human health and climate (Bauer et al., 2007; Ehn et al., 2014; Pye et al., 2010). Particulate nitrate contributes to poor air quality and represents a significant portion of ambient fine particulate matter ($PM_{2.5}$), negatively affecting the human respiratory and cardiovascular systems (Xing et al., 2016). Wet and dry deposition of $tNO_3$ contributes bioavailable nitrogen to often sensitive ecosystems (Galloway et al., 2004; Greaver et al., 2016; Pinder et al., 2012; Walker et al., 2019). In the US, $NO_x$ emissions from power plants and vehicles have dramatically declined over the last several decades due to effective regulations (Hand et al., 2014). Yet, atmospheric nitrogen deposition remains a major terrestrial stressor, which has important implications for land and water quality and interacting effects with climate (Greaver et al., 2016).

Previous studies have suggested that stable nitrogen isotope deltas $\delta(^{15}N) = [R_{sample}(^{15}N/^{14}N)/R_{air-N_2}(^{15}N/^{14}N) - 1]$ may be a powerful observational constraint to enhance our understanding of atmospheric nitrate sources and/or chemical processing (Elliott et al., 2009, 2007; Beyn et al., 2014, 2015; Freyer, 1991; Savard et al., 2017; Savarino et al., 2013; Vicars et al., 2013; Chang et al., 2019; Li et al., 2019; Zong et al., 2017; Hastings et al., 2009; Geng et al., 2014). Precursor $NO_x$ emission sources tend to have distinct $\delta(^{15}N)$ values (or "fingerprints") dependent on formation mechanisms (Miller et al., 2017, 2018; Felix et al., 2012; Walters et al., 2015a, b; Li and Wang, 2008; Yu and Elliott, 2017). For example, biogenic soil emissions tend to have low $\delta(^{15}N, NO_x)$ values of typically less than -25 ‰ (Miller et al., 2018; Yu and Elliott, 2017), stationary liquid fuel combustion has been measured to range between -19.7 to -13.9 ‰ (Walters et al., 2015a), on-road vehicle plumes have been measured to have a range of -9 to -2 ‰ (Miller et al., 2017), and coal combustion tends to have elevated values with a range of 9.8 to 19.8 ‰ (Felix et al., 2012). If these $\delta(^{15}N)$ emission source signatures are proportionally transferred into



atmospheric nitrate, it can be a useful observational constraint for tracking precursor $NO_x$ emission sources to spatiotemporal deposition patterns (Hastings et al., 2013). However, chemical and physical processing associated with $NO_x$ cycling and
formation of atmospheric nitrate can also induce significant isotope fractionation, such that $\delta(^{15}N)$ may not be conserved from emission to deposition (Freyer, 1991; Freyer et al., 1993; Walters et al., 2016; Walters and Michalski, 2015a; Li et al., 2020; Walters and Michalski, 2016a; Vicars et al., 2013). These $\delta(^{15}N)$ fractionations are associated with equilibrium isotope effects (EIE), unidirectional kinetic isotope effects (KIE), and photo-induced fractionation isotope effects (PHIE) (Freyer, 1991; Freyer et al., 1993; Walters et al., 2016; Walters and Michalski, 2015a; Li et al., 2020; Walters and Michalski, 2016a; Michalski
et al., 2020). Accounting for these isotope effects has been shown to be important to understand for $\delta(^{15}N)$ to be used as quantitative tracker of precursor emission sources and chemical effects (Li et al., 2020; Vicars et al., 2013; Michalski et al., 2020; Walters et al., 2018; Savarino et al., 2013; Chang et al., 2018, 2019; Feng et al., 2020).

The northeastern US remains important to monitor due to its high population density, transport patterns, historically degraded
air quality, and elevated acid deposition influenced by $NO_x$ emissions and transformations (Sickles and Shadwick, 2015). Previous landmark $\delta(^{15}N)$ studies of atmospheric nitrate in this region have reported significant correlations between concentration and $\delta(^{15}N)$ of atmospheric nitrate in wet (National Atmospheric Deposition Program; NADP) and dry deposition (Clean Air Status and Trends Network; CASTNET) samples with regional stationary $NO_x$ emission sources from power plant and industrial sectors in the mid-2000s (Elliott et al., 2007, 2009). Considering dramatic $NO_x$ emission changes over the past
decades, it is critical to update our understanding of atmospheric $tNO_3$ deposition's precursor sources and drivers in polluted regions such as the northeastern US. Furthermore, our understanding of $\delta(^{15}N, NO_x)$ emission signatures and $\delta(^{15}N)$ isotope fractionation patterns has significantly improved in recent years. In this study, we have measured the $\delta(^{15}N)$ compositions of $HNO_3$ and $pNO_3$ from CASTNET samples collected in the northeastern US from December 2016 to 2018. Our study contributes to an update on the spatiotemporal $\delta(^{15}N)$ compositions and interpretation of atmospheric $tNO_3$ in the northeastern
US and our understanding of the concentration and $\delta(^{15}N)$ drivers of atmospheric $tNO_3$ after a period of aggressive $NO_x$ emission reductions.

## 2 Materials and Methods

### 2.1 CASTNET Filter Samples

Filter samples from December 2016 to 2018 were obtained from the US EPA CASTNET program for several sites in the
northeastern US, including, (from West to East) Connecticut Hill, NY (CTH110; 42.40° N, -76.65° W), Abington, CT (ABT147; 41.84° N, -72.01° W) and Woodstock, NH (WST109; 43.94° N, -71.70° W) (Figure 1). CASTNET is a national monitoring program sponsored by the US EPA to assess spatiotemporal trends in pollutant concentrations and atmospheric deposition. The CASTNET monitoring locations have been sited to avoid the influence of major cities, highways, local activities, and point source pollution and are expected to be regionally representative (Clarke et al., 1997). As part of this





program, filter pack samples were collected at one-week intervals and included a series of Teflon and Nylon filters for the separate collection of $pNO_3$ and $HNO_3$. Following standard protocols, these filter samples were extracted, measured for concentrations, and stored in an EPA laboratory at room temperature for up to two years until shipment to Brown University.

The filter extracts were re-measured for the total concentrations of nitrate ($NO_3^-$) and nitrite ($NO_2^-$) utilizing standard
colorimetric methods (i.e., US EPA Method 353.2) on an automated discrete UV-Vis Analyzer (SmartChem Westco Scientific Instruments, Inc.) at Brown University. The detection limit was 0.1 and 0.3 μM for $NO_2^-$ and $NO_3^-$, respectively, and the pooled relative standard deviation of replicate quality control standards was better than 3 %. The nitrate concentrations reported by CASTNET were compared to our measured concentrations and gave a near 1:1 relationship for all sites and both filter types, indicating excellent $NO_3^-$ stability in the filter extracts (Figure 2). Equal volumes of the extracted samples were
combined into approximately monthly aggregates to provide sub-seasonal resolution of nitrogen isotope analysis for $HNO_3$ and $pNO_3$. For samples where $[NO_2^-] > 0.1$ μM, $NO_2^-$ was removed using a sulfamic acid treatment (Granger and Sigman, 2009), as the presence of $NO_2^-$ will cause interference when measuring the nitrogen and oxygen isotope ratios of the nitrate (see below). The samples were then frozen until subsequent isotopic analysis.

**2.2 Isotopic Analysis**

Nitrogen stable isotopic analysis was conducted for $HNO_3$ and $pNO_3$ from the monthly aggregated filter extracts using the well-established bacterial denitrifier method (Sigman et al., 2001; Casciotti et al., 2002). Briefly, samples were injected into vials containing P. aureofaciens that quantitatively convert $NO_3^-$ (and $NO_2^-$) to nitrous oxide ($N_2O$). The generated $N_2O$ was concentrated and purified using an automatic purge and trap system and introduced to a continuous flow isotope ratio mass
spectrometer with a modified gas bench interface at Brown University. Measurement of $N_2O$ was conducted at $m/z$ of 44, 45, and 46 to determine $\delta(^{15}N)$, and unknowns were corrected relative to internationally recognized nitrate salt reference materials that included: USGS34 ($\delta(^{15}N) = -1.8$ ‰), USGS35 ($\delta(^{15}N) = 2.7$ ‰), and IAEA-N3 ($\delta(^{15}N) = 4.7$ ‰) (Böhlke et al., 1993; Böhlke et al., 2003). Isobaric influences from $^{17}O$ contributions were corrected based on a separate analysis, in which $N_2O$ was thermally decomposed to $O_2$ by passing through a gold tube heated to 770 ℃ and then measured at $m/z$ 32, 33, and 34 for
$\Delta(^{17}O)$ (defined as: $\Delta(^{17}O) = \delta(^{17}O) - 0.52 \times \delta(^{18}O)$) determination (Kaiser et al., 2007). This correction resulted in a $\delta(^{15}N)$ decrease typically near 1.5 ‰. All isotopic reference materials were diluted to similar concentrations as samples and run intermittently in each batch analysis. The overall standard deviations of isotopic reference materials were $\sigma(\delta(^{15}N)) = 0.2$ ‰ ($n=13$), 0.4 ‰ ($n=13$), and 0.2 ‰ ($n=15$) for USGS34, USGS35, and IAEA-N3, respectively.



## 2.3 HYSPLIT Modeling and 'Openair' Package

Air mass back-trajectories were computed using the HYSPLIT model and the North American Regional Reanalysis (NARR) 12 km dataset (Stein et al., 2015). 72-hour back trajectories were calculated at 50 m above ground level every other day for each site (CTH110, ABT147 and WST109) across the sample collection period from December 2016 to 2018. The trajectory data was collated with the reported CASTNET concentration data ($pNO_3$, $HNO_3$, and $tNO_3$) at a weekly resolution to link concentration trends to the source regions for nitrate. Using the 'openair' program package in R (Carslaw and Ropkins, 2012), geospatial statistical analysis that included back-trajectory clustering and the concentrated weighted trajectory (CWT) algorithm was conducted to determine patterns of transport and major contributing source regions for atmospheric nitrate. The CWT model is a statistical tool that utilizes the air mass residence time analysis to identify emission source regions (Hsu et al., 2003; Salamalikis et al., 2015; Cheng et al., 2013; Dimitriou et al., 2015). For each grid cell, CWT calculates the concentration of a pollutant as the following (1):

$$\bar{c}_{ij} = \frac{1}{\sum_{k=1}^{N} \tau_{ijk}} \sum_{k=1}^{N} c_k \tau_{ijk} \tag{1}$$

where $i$ and $j$ are the indices of grid, $k$ is the index of trajectory, $N$ is the total number of trajectories used in the analysis, $c_k$ is the pollutant concentration measured upon arrival of trajectory $k$, and $t_{ijk}$ is the residence time of trajectory $k$ in grid cell $(i,j)$. A high value of $\bar{c}_{ij}$ means that air parcels that pass over the cell $(i,j)$ would, on average, cause a high concentration at the receptor site (Carslaw and Ropkins, 2012).

## 2.4 NO$_x$ Emissions Database and δ$^{15}$N(NO$_x$) Estimation

Monthly anthropogenic NO$_x$ emission density estimates were extracted from a recent sector and fuel-based emission inventory to understand how precursor NO$_x$ emissions contribute to nitrate concentration and isotope trends (McDuffie et al., 2020). The monthly NO$_x$ emissions data were reported in gridded 0.5º × 0.5º units divided into eleven anthropogenic sectors: Agriculture, Energy Production, Industry, On-Road Transportation, Non-Road Transportation, Combustion-Residential, Combustion-Commercial, Combustion-Other, Shipping, Solvents, and Waste. (Note that solvents are not a source of NO$_x$ emissions.) The combustion sector emissions were further broken down into fuel types (coal, solid biofuel, and liquid fuel), while non-combustion emissions were assigned to a single "process" fuel type. Monthly NO$_x$ emission density estimates by sector and fuel-type data were extracted from the nitrate source regions determined from the CWT analysis. The regions were defined using spatial polygons in 'R', which sets latitude and longitude coordinates to retrieve spatially encoded data. Monthly $\delta(^{15}N, NO_x)$ was modeled based on isotope mass-balance using the fraction of NO$_x$ emissions by sector and fuel type and previously reported $\delta(^{15}N, NO_x)$ emission signatures following a previously described method (2) (Walters et al., 2015a):

$$\delta(^{15}N, \ NO_x) = \sum_{i=1}^{n} f_i \ \delta_i(^{15}N, NO_x) \tag{2}$$

where $\delta_i$ is the emission signature of source and $f_i$ is the fraction contributing to the NO$_x$ emissions. The considered $\delta(^{15}N, NO_x)$ emission signatures included grouped agriculture/waste (Miller et al., 2018), on-road transportation (Miller et al., 2017),





non-road transportation (Walters et al., 2015a), and shipping (Walters et al., 2015a). Energy production, industry, and combustion were grouped by fuel type as either Combustion – coal & solid biofuel (Felix et al., 2012) or Combustion – liquid fuel & process (Walters et al., 2015a). The emission inventory only considers anthropogenic $NO_x$ emissions such that natural

emissions such as lightning and wildfires were not considered. Table 1 summarizes the $\delta(^{15}N, NO_x)$ emission signatures (Walters et al., 2015a; Miller et al., 2018, 2017; Felix et al., 2012).

## 2.5 GEOS-Chem Simulations

The GEOS-Chem global model of atmospheric chemistry (www.geos-chem.org) was utilized to predict $NO_x$ and $O_3$

concentrations in the regions of the various CASTNET sites to account for $\delta(^{15}N)$ isotope fractionation that occur during chemical reactions. We use version 13.2.1 (doi:10.5281/zenodo.5500717) of the model driven by GEOS5-FP assimilated meteorology from the NASA Global Modeling and Assimilation Office (GMAO). A nested grid (0.25° latitude × 0.3125° longitude horizontal resolution; 25 km) simulation was conducted over the northeastern United States (97°-60° W; 35°-60° N) in 2017 and 2018. Boundary conditions were from global simulations performed at 4° latitude × 5° longitude horizontal

resolution for the same years after a one-year initialization. Gas- and aerosol-phase chemistry was simulated using the default "fullchem" mechanism (Bates and Jacob, 2019; Wang et al., 2021). Inorganic gas and aerosol partitioning were conducted using version 2.2 of the ISORROPIA II thermodynamic equilibrium model (Fountoukis and Nenes, 2007). All default anthropogenic emissions were applied, which is primarily version 2.0 of the Community Emissions Data System (Hoesly et al., 2018) as previously implemented (McDuffie et al., 2020). Natural emissions respond to local meteorology and include

biogenic VOCs from terrestrial plants and the ocean (Millet et al., 2010; Guenther et al., 2012; Hu et al., 2015; Breider et al., 2017), $NO_x$ from lightning and soil microbial activity (Murray et al., 2012; Hudman et al., 2012), mineral dust (Ridley et al., 2012), and sea salt (Jaeglé et al., 2011; Huang and Jaeglé, 2017). Biomass burning emissions were monthly means from version 4.1s of the Global Fire Emissions Database (GFED4.1s; (van der Werf et al., 2017). Wet deposition for water-soluble aerosols is described by Liu et al., 2001 and by Amos et al., 2012 for gases. Dry deposition is based on the resistance-in-series

scheme of Wesely and Lesht, 1989.

## 3. Results and Discussion

### 3.1 Atmospheric Nitrate Spatiotemporal Concentrations

The atmospheric nitrate concentrations are shown in Figure 3 and summarized in Table 2. Overall, the mean concentrations

of the three examined Northeastern US CASTNET sites were significantly different but showed similar seasonal trends. Across the sites, the annual concentrations of $HNO_3$, $pNO_3$, and $tNO_3$ were significantly higher at Abington, CT and Connecticut Hill, NY than at Woodstock, NH ($p < 0.0001$). The concentrations were binned by season including Winter (DJF),





Spring (MAM), Summer (JJA), and Autumn (SON), which indicated seasonal statistical differences at the considered CASTNET sites (Figure 4). The HNO$_3$ concentrations were significantly greater during the winter for Woodstock, NH, than in other seasons ($p<0.01$).  Additionally, HNO$_3$ at Abington, CT, was significantly higher during summer than in autumn ($p<0.001$).  There was no significant seasonal difference in HNO$_3$ concentrations at Connecticut Hill, NY.  At all three sites, the concentrations of pNO$_3$ were greatest during the winter and lowest during the summer.  These findings were consistent with previous reports of CASTNET samples in the Northeastern and Midwestern US collected from 2004 to 2005, in which pNO$_3$ concentrations were highest in the winter and lowest in the summer and with little seasonal variation in HNO$_3$ (Elliott et al., 2009).  Thus, even as NO$_x$ emissions have been dramatically decreasing in the Northeastern US over the past decade, the HNO$_3$ and pNO$_3$ seasonal trends have been retained.

Clustered air mass back trajectories were calculated for the CASTNET sites (Figure 5).  The annual clustered trajectories indicate that most air masses were associated with westerlies with prevailing winds from the continental US and Canada for all the considered CASTNET sites.  The clustered trajectories also indicate the influence of marine/coastal air masses and winds from the northeast.  The CWT analysis of tNO$_3$ concentrations indicated that contributing source regions tended to be within approximately 1000 km from the CASTNET sites (Figure 5).  Like the cluster trajectory results, the CWT analysis indicated that the tNO$_3$ source contributing regions tended to extend towards the west and northwest of the CASTNET sites with minimal contributions east of the sites.  Similar source regions were identified for the various CASTNET sites, but there were slight spatial differences due to the location of the sites.  For example, the source regions contributing to CTH110 tended to extend further from the Midwest compared to the other sites, and a higher relative contribution from southeast Canada was identified for the WST109 site.

## 3.2 Atmospheric Nitrate Spatiotemporal $\delta(^{15}N)$ Compositions

The measured atmospheric nitrate $\delta(^{15}N)$ data are shown in Figure 6 and summarized in Table 2.  The $\delta(^{15}N)$ data indicated significant spatial differences but with consistent seasonal patterns for $\delta(^{15}N, HNO_3)$, $\delta(^{15}N, pNO_3)$, and $\delta(^{15}N, tNO_3)$.  The $\delta(^{15}N)$ values were highest for Abington, CT, second highest for Connecticut Hill, NY and lowest for Woodstock, NH. Across the sites, there was a consistent offset between $\delta(^{15}N, HNO_3)$ and $\delta(^{15}N, pNO_3)$, in which $\delta(^{15}N, pNO_3)$ tends to have higher values relative to $\delta(^{15}N, HNO_3)$ that averaged a ($3.9 \pm 1.8$) ‰ ($n=79$) difference for simultaneously collected samples. This value was in close agreement with the theoretical isotope effect associated with N isotopic equilibrium between NO$_3^-$ and HNO$_3$, which has been calculated to be 3.2 ‰ at 298 K, favoring the preferential partitioning of $^{15}N$ into NO$_3^-$ (Walters and Michalski, 2015b).

Across all sites, $\delta(^{15}N, HNO_3)$, $\delta(^{15}N, pNO_3)$, and $\delta(^{15}N, tNO_3)$ indicated consistent temporal patterns, with the highest values observed during the winter and lowest values during the summer (Figure 7).  These findings were similar to previous $\delta(^{15}N)$





measurements from $HNO_3$, $pNO_3$, and precipitation $NO_3^-$ samples collected in the early 2000s in the Midwestern and Northeastern US, which also reported a significant spatiotemporal variation (Elliott et al., 2009, 2007). The CTH110 site was previously analyzed for its $\delta(^{15}N)$ deltas in the early 2000s (Elliott et al., 2009). Overall, the range of measured $\delta(^{15}N)$ at CTH110 was lower in 2017-2018 ($\delta(^{15}N, HNO_3)$ = -11.1 ‰ to -0.1 ‰; $\delta(^{15}N, pNO_3)$ = -6.8 ‰ to 4.4 ‰), compared to

measurements conducted for 2004-2005 ($\delta(^{15}N, HNO_3)$ = -5 ‰ to 10 ‰; $\delta(^{15}N, pNO_3)$ = -1.0 ‰ to 12 ‰) (Elliott et al., 2009). This trend is consistent with an expected decrease in $\delta(^{15}N)$ of atmospheric nitrate following implementation of $NO_x$ reduction technologies on electricity generation units and their subsequent relative decrease in $NO_x$ emissions (Felix et al., 2012).

### 3.3 $NO_x$ Emission Modeling

Previous spatiotemporal $\delta(^{15}N)$ differences in atmospheric nitrate in the Midwestern and Northeastern US had been concluded to reflect the importance of precursor emission sources (Elliott et al., 2009, 2007). Specifically, stationary source $NO_x$ emissions associated with coal combustion with a high $\delta(^{15}N, NO_x)$ emission signature were suggested to drive higher $\delta(^{15}N)$ values during winter and a longitudinal gradient across the Midwestern and Northeastern US (Elliott et al., 2009). To test this hypothesis on the current dataset, the monthly $NO_x$ emission densities speciated by sector and fuel-specific sources (McDuffie

et al., 2020) were extracted for spatial polygons that approximately corresponded to the identified $tNO_3$ source contributing regions from the CWT analysis (Figure 5). Across all sites, the predicted $NO_x$ emission densities indicated similar seasonal variability, with a maximum observed during winter from higher residential, commercial, and other combustion emissions. This increase is due to a significant heating demand during this period (Figure 8). A local maximum was also observed during summer due to increased emissions related to electricity generation for cooling. The absolute $NO_x$ emission densities across

sites were broadly consistent with the $tNO_3$ concentration trends, in which higher $NO_x$ emission and $tNO_3$ were observed for CTH110 and ABT147 compared to WST109. Across the sites, there were similar annual contributing $NO_x$ emission sectors for the identified source regions contributing $tNO_3$ to the study sites that included energy (21.9 %, 22.5 %, 23.5 %), industry (14.4 %, 14.6 %, 14.1 %), non-road transport (17.3 %, 16.2 %, 15.0 %), combustion-residential, commercial, other (12.8 %, 14.2 %, 14.3 %), road (23.9 %, 23.2 %, 23.3 %), shipping (7.5 %, 7.5 %, 8.5 %), and agricultural/waste (2.1 %, 1.7 %, 1.5 %)

for CTH110, ABT147 and WST109, respectively. Additionally, there were similar annual $NO_x$ emission density contributing fuel-types across sites including Biofuel (2.6 %, 2.7 %, 2.7 %), Coal (5.8 %, 5.2 %, 4.8 %), Liquid-fuel (76.4 %, 75.0 %, 73.9 %), and Process-based emissions (15.3 %, 17.2 %, 18.7 %) for the identified source regions contributing to $tNO_3$ at CTH110, ABT147, and WST109, respectively.

The monthly $\delta(^{15}N, NO_x)$ was calculated using the $NO_x$ emission estimates, assumed emission source values, and isotope mass balance (Figure 8). Overall, this calculation indicated limited spatial variability with an annual $\delta(^{15}N, NO_x)$ average of (-11.7±0.1) ‰, (-11.6±0.1) ‰, and (-11.8±0.8) ‰ for ABT147, CTH110, and WST109, respectively. We note that for each of the monthly $\delta(^{15}N, NO_x)$ estimations, the propagated uncertainty based on the $\delta(^{15}N, NO_x)$ emission signature reported





uncertainty was approximately ±3.4 ‰ and was not seasonally variable.    There was limited seasonality in the modeled $\delta(^{15}N,$ $NO_x$) across all sites that was different by no more than 0.3 ‰ in the monthly mean values.  The highest modeled mean $\delta(^{15}N,$ $NO_x$) values occurred during the summer due to increased emissions from the energy production sector; namely, an increase in coal and solid biofuel combustion, which has an elevated $\delta(^{15}N, NO_x)$ signature (Table 1) (Felix et al., 2012).

The modeled $\delta(^{15}N, NO_x)$ was compared with the measured monthly $\delta(^{15}N, tNO_3)$ to remove the potential $\delta(^{15}N)$ phase fractionation between $HNO_3$ and $pNO_3$. Overall, the modeled $\delta(^{15}N, NO_x)$ was lower than the observed $\delta(^{15}N, tNO_3)$ values, and the lack of spatiotemporal variability in the modeled $\delta(^{15}N, NO_x)$ was in direct contrast to the $\delta(^{15}N, tNO_3)$ values (Figure 8).    This finding suggests that seasonal changes in $NO_x$ emission sectors by fuel type did not drive significant seasonal variability in $\delta(^{15}N, NO_x)$ or $\delta(^{15}N, tNO_3)$ across the considered CASTNET sites.  Previous studies of atmospheric nitrate in the northeastern/midwestern US during the early 2000s found that stationary source $NO_x$ emissions, including power plants
and industrial processes, were strongly correlated with $\delta(^{15}N, NO_3^-)$ (Elliott et al., 2009, 2007), which is inconsistent with our results from a similar region from samples collected 10 years later.  This inconsistency may suggest that the dramatic decrease in stationary combustion emissions, particularly from coal combustion, has led to decoupling between $NO_x$ emissions and $\delta(^{15}N)$ of atmospheric nitrate.

The mismatch between the modeled $\delta(^{15}N, NO_x)$ and the observed $\delta(^{15}N, tNO_3)$ did not suggest that there were significant inaccuracies in the $NO_x$ emission inventories, such as under constrained soil emissions and/or not accounting for natural sources of $NO_x$ such as lightning.  Soil $NO_x$ emissions have a characteristic low $\delta(^{15}N, NO_x)$ emission signature (Miller et al., 2018; Yu and Elliott, 2017), such that underestimation of soil emissions could not explain the observed mismatch as the modeled $\delta(^{15}N, NO_x)$ was already lower than the observed $\delta(^{15}N, tNO_3)$.  Lightning-generated $NO_x$ was also unlikely to explain
the model mismatch with observations.  Lightning $NO_x$ has a reported $\delta(^{15}N)$ signature near 0 ‰ (Hoering, 1957), such that to match the modeled $\delta(^{15}N, NO_x)$ with the observed $\delta(^{15}N, tNO_3)$ would require a substantial amount of lightning-produced $NO_x$. However, lightning $NO_x$ emissions are excepted to be several times smaller than $NO_x$ emissions from anthropogenic sources (Murray, 2016).  Thus, we next considered if the spatiotemporal $\delta(^{15}N, tNO_3)$ variability observed at the CASTNET sites during 2016-2018 can be explained by $\delta(^{15}N)$ isotope fractionation associated with $NO_x$ oxidation.


### 3.4 $NO_x$ Cycle Isotope Fractionation

$NO_x$ oxidation to atmospheric nitrate has been suggested to induce significant $\delta(^{15}N)$ fractionation associated with $NO_x$ cycling and the reaction pathways leading to nitrate formation (Walters and Michalski, 2015b; Freyer, 1991; Freyer et al., 1993; Walters et al., 2016; Walters and Michalski, 2016b; Michalski et al., 2020; Li et al., 2020).    We calculated the influence of
$\delta(^{15}N)$ fractionation associated with $NO_x$ cycling on $\delta(^{15}N, NO_2)$ derived from previous studies as the following (3):

$$\delta(^{15}N, \ NO_2) \approx \delta(^{15}N, \ NO_x) + {}^{15}\varepsilon(NO_2/NO) \times (1 - f(NO_2)) \tag{3}$$





where $\delta(^{15}N, NO_x)$ represents the modeled emissions (Figure 8), $^{15}\varepsilon(NO_2/NO)$ is the isotope effect associated with NO conversion to $NO_2$, and $f(NO_2)$ represents the amount fraction of $NO_2$ in $NO_x$ (i.e., $f(NO_2) = [NO_2]/[NO_x]$). The $^{15}\varepsilon(NO_2/NO)$ value represents a combination of the $NO_x$ equilibrium isotope effect (EIE) and the Leighton Cycle isotope effect (LCIE)

(Freyer et al., 1993; Walters et al., 2016; Li et al., 2020). Briefly, the EIE between NO and $NO_2$ has been shown to have an isotope effect of (28.9±1.9) ‰ from an experimental investigation under ambient $NO_x$ conditions (Li et al., 2020). The effect favors higher $\delta(^{15}N)$ values in $NO_2$, which dominates $\delta(^{15}N, NO_x)$ fractionation during conditions of high $NO_x$ concentrations (Freyer et al., 1993; Walters et al., 2016; Li et al., 2020). The LCIE represents a combination of the kinetic isotope effect associated with NO oxidation, primarily driven by reaction with $O_3$, and the isotope effect associated with $NO_2$ photolysis

(Walters et al., 2016; Li et al., 2020). The dominant factor in LCIE is likely the NO + $O_3$ fractionation, as the $NO_2$ photolysis isotope effect has been suggested to have a near negligible fractionation (Michalski et al., 2020). Indeed, laboratory investigation of the LCIE suggests an enrichment value of (-10±5) ‰, which is in close agreement with the KIE from *ab initio* calculations of NO + $O_3$ of -6.7 ‰ at 298 K (Walters and Michalski, 2016a). In contrast to the EIE, the LCIE dominants $NO_x$ $\delta(^{15}N)$ fractionation during conditions of low $NO_x$ concentrations (Li et al., 2020).


We have estimated the relative role of EIE and LCIE based on the following (4):

$$^{15}\varepsilon(NO_2/NO) \ = f_{EIE}\left(^{15}\varepsilon_{EIE}\right) + (1 - f_{EIE})\left(^{15}\varepsilon_{LCIE}\right) \tag{4}$$

The $f_{EIE}$ represents the relative rate of $NO_x$ EIE to NO oxidation and is calculated as the following (5):

$$f_{EIE} = \frac{k(NO_x - EIE)[NO_2]}{k(NO+O_3)[O_3] + k(NO_x - EIE)[NO_2]} \tag{5}$$

where $k(NO_x\text{-EIE})$ is the reaction rate of $NO_x$ EIE with a reported value of $8.14\times10^{-14}$ cm$^3$ s$^{-1}$ (Sharma et al., 1970) and $k(NO+O_3)$ is the NO + $O_3$ reaction rate of $1.73\times10^{-14}$ cm$^3$ s$^{-1}$. (Atkinson et al., 2004).

The value of $f_{EIE}$ was calculated using modeled NO, $NO_2$, and $O_3$ concentrations from GEOS-Chem integrated over the source regions that contributed $tNO_3$ to the CASTNET sites. The modeled $O_3$ and $NO_x$ concentrations indicated opposite seasonal

trends for all considered source regions: $O_3$ reached a maximum during summer due to increased photochemical activity, while $NO_x$ reached a maximum during winter due to lower photolysis frequencies and relatively higher $NO_x$ emissions, as expected (Figure 9). The modeled $f(NO_2)$ closely followed the $O_3$ seasonal profile (Figure 9). The calculated $f_{EIE}$ also followed the $NO_x$ seasonal profile with peaks during the winter and ranged from 0.124 to 0.513 across the CASTNET sites (Figure 9), which is the expected trend as the influence of EIE on $\delta(^{15}N)$ fractionation is highest during conditions of higher $NO_x$

concentrations relative to $O_3$ (Freyer et al., 1993; Walters et al., 2016; Li et al., 2020). The $f_{EIE}$ averaged 0.255±0.108, 0.271±0.115, 0.218±0.093 for ABT147, CTH110, and WST109, indicating that $\delta(^{15}N)$ fractionation was largely driven by the NO + $O_3$ oxidation rather than by $NO_x$ EIE due to the low modeled $NO_x$ concentration relative to $O_3$. The calculated $^{15}\varepsilon(NO_2/NO)$ had a similar seasonal profile as $f_{EIE}$, with peaks during the winter compared with summer and ranged from -5.2





to 10.0 ‰ across the CASTNET sites with an average of (0.5±4.5) ‰, (-0.1±4.2) ‰, and (-1.5±3.6) ‰ for CTH110, ABT147,

and WST109, respectively (Figure 9).

The $\delta(^{15}N, NO_2)$ was then calculated using the monthly calculated $^{15}\varepsilon(NO_2/NO)$, modeled $f(NO_2)$, and modeled $\delta(^{15}N, NO_x)$.
Overall, the $\delta(^{15}N, NO_2)$ ranged from -12.4 to -10.3 ‰ across the CASTNET sites and averaged (-11.5±0.5) ‰, (-11.7±0.5)
‰, and (-12.0±0.4) ‰ for CTH110, ABT147, and WST109, respectively (Figure 9). These annual averages were nearly

identical to the modeled $\delta(^{15}N, NO_x)$ values. There was slight seasonal variability in the calculated $\delta(^{15}N, NO_2)$, with slightly
higher values during winter than in summer. However, neither the magnitude of the seasonal variability, which was no more
than 1.6 ‰ nor the absolute value of the calculated $\delta(^{15}N, NO_2)$ agreed with the measured $\delta(^{15}N, tNO_3)$. Overall, this indicates
that $\delta(^{15}N)$ fractionation associated with $NO_x$ cycling played an insignificant role in explaining the spatiotemporal variabilities
observed for $\delta(^{15}N, tNO_3)$ at the CASTNET sites.


### 3.5  Nitrate Formation Isotope Fractionation

Nitrogen isotope fractionation has also been suggested to occur during reactions leading to $HNO_3$ and/or $pNO_3$ formation
(Walters and Michalski, 2015b, 2016b; Michalski et al., 2020). Assuming atmospheric nitrate formation represents an
irreversible reaction in an open system with a constant supply of $NO_x$ emissions, we model the $\delta(^{15}N, tNO_3)$ as the following

325    (6):

$$\delta(^{15}N, tNO_3) = \delta(^{15}N, NO_2) + {}^{15}\varepsilon(tNO_3/NO_2) \tag{6}$$

The $^{15}\varepsilon(tNO_3/NO_2)$ corresponds to the enrichment factor associated with converting $NO_2$ to $tNO_3$. We calculated the
$^{15}\varepsilon(tNO_3/NO_2)$ as the difference between the measured $\delta(^{15}N, tNO_3)$ and the calculated $\delta(^{15}N, NO_2)$ (Figure 10). Across all
sites, $^{15}\varepsilon_{calc}(tNO_3/NO_2)$ ranged from 1.6 to 16.1 ‰, with an average of (8.7±3.8) ‰, (10.9±3.5) ‰, and (6.9±2.9) ‰, for

CTH110, ABT147, and WST109. Additionally, the $^{15}\varepsilon_{calc}(tNO_3/NO_2)$ indicated strong seasonality with higher values during
the winter compared to the summer. The shift in the seasonal $^{15}\varepsilon_{calc}(tNO_3/NO_2)$ were likely attributed to a change in nitrate
formation pathway, as previously suggested (Li et al., 2021).

The two dominant polluted mid-latitude nitrate formation pathways include $NO_2$ oxidation via hydroxyl radical (R1) and $N_2O_5$

hydrolysis (R2):

$$NO_2 + OH + M \longrightarrow HNO_3 + M \tag{R1}$$

$$N_2O_5 + H_2O(surface) \longrightarrow 2HNO_3 \tag{R2}$$

These reactions have an isotope effect of -3 ‰ based on the reduced masses of the transition complex (Freyer, 1991) and 25.5
‰ at 300 K based on EIE between $NO_2$ and $N_2O_5$ (Walters and Michalski, 2016b) for R1 and R2, respectively, indicating that

the range of the $^{15}\varepsilon_{calc}(tNO_3/NO_2)$ is between these end member values. We estimated the relative role of R1 and R2
contributing to nitrate formation across the considered CASTNET sites based on the following (7):





$$^{15}\varepsilon_{\text{calc}}(\text{tNO}_3/\text{NO}_2) = f(\text{NO}_2 + \text{OH}) \times (^{15}\varepsilon(\text{NO}_2 + \text{OH}) + (1 - f(\text{NO}_2 + \text{OH})) \times (^{15}\varepsilon(\text{N}_2\text{O}_5, \text{T})) \tag{7}$$

assuming that R1 and R2 dominate the observed tNO$_3$ formation, as expected for the polluted mid-latitudes (Alexander et al., 2020) (8):

$$f(\text{NO}_2 + \text{OH}) + f(\text{N}_2\text{O}_5) = 1 \tag{8}$$

where $^{15}\varepsilon_{\text{calc}}(\text{tNO}_3/\text{NO}_2)$ is our calculated results (Figure 10), $f(\text{NO}_2+\text{OH})$ and $f(\text{N}_2\text{O}_5)$ correspond to the fractional contribution of R1 and R2, respectively, $^{15}\varepsilon(\text{NO}_2+\text{OH})$ = -3 ‰ (Freyer, 1991a), and $^{15}\varepsilon(\text{N}_2\text{O}_5, \text{T})$/‰ = -0.163*T/K+74.08 for a temperature range of 250 to 305 K (Walters and Michalski, 2016b). We utilized the temperature derived over the source regions contributing to the CASTNET sites from the GEOS-Chem simulations in our calculations, which indicated a range in the monthly temperature of 262.4 to 294.8 K, corresponding to a range in $^{15}\varepsilon(\text{N}_2\text{O}_5, \text{T})$/‰ of 26.4 to 31.3 ‰. Overall, we estimated $f(\text{NO}_2+\text{OH})/f(\text{N}_2\text{O}_5)$ contributed 0.63±0.11/0.37±0.11, 0.56±0.09/0.44±0.09, and 0.69±0.8/0.31±0.08 to CTH110, ABT147, and WST109, respectively (Figure 10). For each of the considered sites, the calculated $f(\text{NO}_2+\text{OH})$ peaked during the summer and $f(\text{N}_2\text{O}_5)$ peaked during the winter, consistent with expected seasonal atmospheric nitrate formation and model results (Alexander et al., 2020). We acknowledge that are uncertainties in our model regarding potential contributions from other nitrate formation pathways and the considered enrichment factors that have not been experimentally determined. Nevertheless, our results highlight that seasonal $\delta(^{15}\text{N}, \text{tNO}_3)$ values were driven by nitrate formation based on our current understanding of fractionation patterns.

## 4. Conclusions

Significant spatiotemporal differences in concentrations and $\delta(^{15}\text{N})$ were observed for atmospheric nitrate in the northesatern US from December 2016 to 2018 from CASTNET locations. These findings were consistent with a previous study of atmospheric nitrate from CASTNET sites collected in the early 2000s, indicating that even after dramatic reductions in NO$_x$ emissions in the Northeastern US over the past decade, atmospheric nitrate spatiotemporal trends have been retained. We focused on evaluating the drivers of the spatiotemporal trends of $\delta(^{15}\text{N})$ observed at the CASTNET sites. Back trajectory and geospatial statistical analyses indicated that atmospheric nitrate source regions tended to be within 1000 km and tended to extend towards the west/northwest of the CASTNET sites. Utilizing NO$_x$ emission data for the identified source regions, we modeled $\delta(^{15}\text{N}, \text{NO}_x)$ for each of the CASTNET sites, indicating no significant spatiotemporal differences. This finding suggested that NO$_x$ emissions were not a key driver of the observed spatiotemporal $\delta(^{15}\text{N})$ variability as previously reported for CASTNET sites in the early 2000s. Instead, we found that $\delta(^{15}\text{N})$ fractionation primarily associated with nitrate formation was the key driver of the observed spatiotemporal $\delta(^{15}\text{N})$ variabilities.

Our results highlight that $\delta(^{15}\text{N})$ of atmospheric nitrate fractionation could lead to new insights via tracking nitrate formation mechanisms. The $\delta(^{15}\text{N})$ fractionation associated with NO$_x$ conversion to atmospheric nitrate reflected the nitrate formation





pathways. Thus, the $\delta(^{15}N)$ of atmospheric nitrate could be a useful way to track the reactions contributing to nitrate formation, similarly to $\Delta(^{17}O)$ (Alexander et al., 2020; Michalski et al., 2003). However, $\delta(^{15}N)$ would arguably be more sensitive to nitrate formation pathways because most of the $\Delta(^{17}O)$ of nitrate reflects $NO_x$ photochemical cycling (NO + $O_3$ vs NO +$RO_2$/$HO_2$) rather than the reactions contributing to nitrate formation. Thus, $\delta(^{15}N)$ and $\Delta(^{17}O)$ could be useful complementary tools to improve our ability to track $NO_x$ oxidation and nitrate formation and compare with model expectations. Future studies are needed to verify the assumed $\delta(^{15}N)$ fractionation values associated with nitrate formation, enabling $\delta(^{15}N)$ to be a useful tool for tracking oxidation chemistry pathways.

**Data Availability.** Data presented in this article are available on the Harvard Dataverse at https://doi.org/10.7910/DVN/X6BB1I and the US EPA CASTNET database.

**Author Contributions.** CB, WW, MGH designed the varying aspects of the study. CB and WWW carried out the laboratory measurements. CB conducted the statistical analysis, backtrajectory calculations, and emissions modeling. LTM contributed GEOS-Chem simulations. CB and WWW prepared the article with contributions from all co-authors.

**Acknowledgements.** We thank Ruby Ho for sampling and laboratory assistance. We are grateful to the US EPA CASTNET program and staff for their cooperation in this study and assistance with receiving archived samples for isotopic analysis.

**Competing Interests.** The authors declare no competing financial interest.

**Financial Support.** National Science Foundation (AGS-2002750); Institute at Brown for Environment and Society Seed Grant; Voss Environmental Fellowship from the Institute at Brown for Environment and Society

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





**Table 1. Summary of $\delta(^{15}N, NO_x)$ emission source values.**

| $NO_x$ Emission Source | $\delta(^{15}N, NO_x)$ (‰) (mean ±σ) | Reference |
|---|---|---|
| Agriculture/Waste[*] | -33.0±12.3 | (Miller et al., 2018) |
| On-Road Transport | -4.7±1.7 | (Miller et al., 2017) |
| Non-Road Transport | -16.8±5 | (Walters et al., 2015a) |
| Shipping | -16.8±5 | (Walters et al., 2015a) |
| Combustion- Coal & Solid Biofuel[**] | 13.6±3.9 | (Felix et al., 2012) |
| Combustion – Liquid Fuel & Process[**] | -16.5±1.7 | (Walters et al., 2015a) |

[*]Waste $NO_x$ emissions represented <1 % of total monthly $NO_x$ emissions within each identified nitrate source region and were lumped with agricultural $NO_x$ emissions

[**]Combustion-Residential, Combustion-Commercial, and Combustion-Other were combined (Combustion) and separated by fuel type (i.e., Combustion- Coal & Solid Biofuel & Combustion-Liquid Fuel & Process). The "Process" Combustion emissions were assumed to have a similar $\delta(^{15}N, NO_x)$ value as liquid fuel.






**Table 2.  Statistical summary including minimum (Min), maximum (Max), mean (Mean), standard deviation (SD), and number of counts (N) for concentration and $\delta(^{15}N)$ of HNO$_3$, pNO$_3$, and tNO$_3$ at the CASTNET sites.**

| Descriptive Statistic | HNO$_3$ Concentration ($\mu$g m$^{-3}$) | $\delta(^{15}N)$ (‰) | pNO$_3$ Concentration ($\mu$g m$^{-3}$) | $\delta(^{15}N)$ (‰) | tNO$_3$ Concentration ($\mu$g m$^{-3}$) | $\delta(^{15}N)$ (‰) |
|---|---|---|---|---|---|---|
| **Connecticut Hill, NY** | | | | | | |
| Min | 0.219 | -11.1 | 0.091 | -6.8 | 0.320 | -9.8 |
| Max | 1.203 | -0.1 | 5.033 | 4.4 | 5.474 | 3.0 |
| Mean(SD) | 0.526(0.200) | -4.7(3.2) | 0.735(0.813) | -0.6(3.2) | 1.261(0.832) | -2.7(4.1) |
| N | 105 | 26 | 105 | 26 | 105 | 26 |
| **Abington, CT** | | | | | | |
| Min | 0.138 | -9.5 | 0.142 | -4.3 | 0.488 | -7.5 |
| Max | 1.326 | 4.3 | 3.466 | 5.8 | 4.375 | 5.1 |
| Mean(SD) | 0.600(0.255) | -2.1(4.4) | 0.723(0.582) | 0.6(2.8) | 1.323(0.662) | -0.9(3.9) |
| N | 107 | 27 | 107 | 27 | 107 | 27 |
| **Woodstock, NH** | | | | | | |
| Min | 0.061 | -11.7 | 0.058 | -6.9 | 0.148 | -10.6 |
| Max | 0.721 | -3.4 | 1.213 | 2.3 | 1.934 | -0.4 |
| Mean(SD) | 0.218(0.094) | -6.7(2.4) | 0.199(0.183) | -1.8(2.7) | 0.417(0.252) | -4.8(3.0) |
| N | 105 | 26 | 105 | 26 | 105 | 26 |






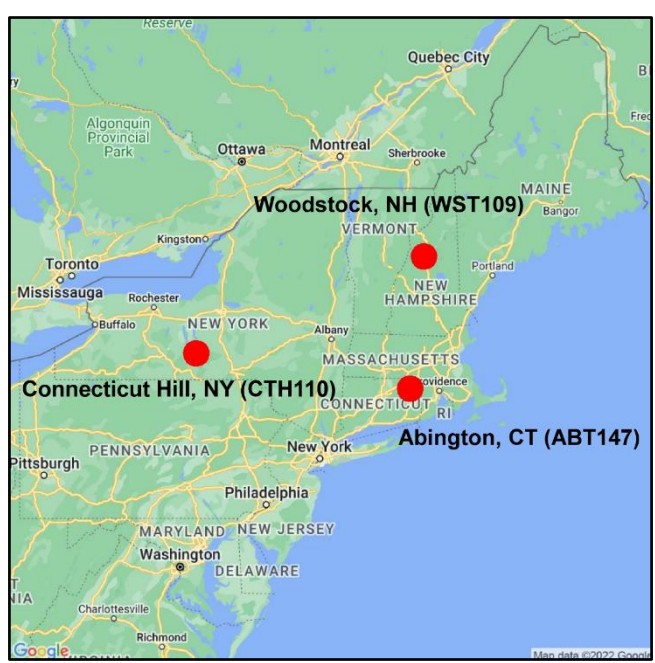

**Figure 1. Map of the three Northeastern CASTNET monitoring sites included in the study. The image was created using Google Maps (Map data ©2022 Google).**









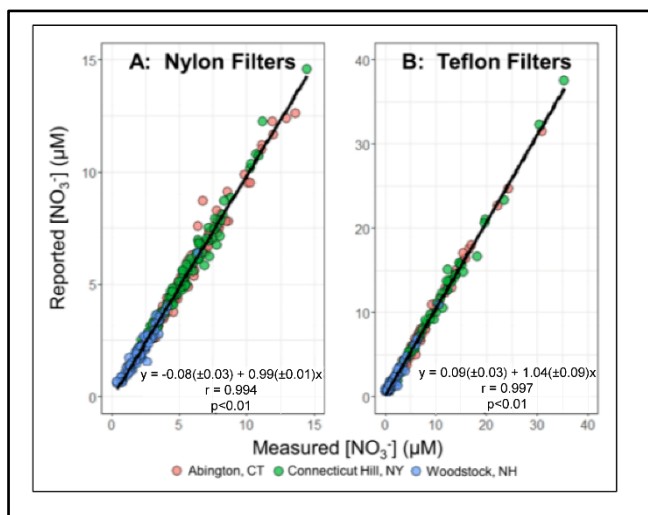

**Figure 2.** Comparison between the nitrate ($NO_3^-$) concentrations reported by CASTNET with those measured at Brown University for the Nylon filter (A) and Teflon Filter (B) extracts.









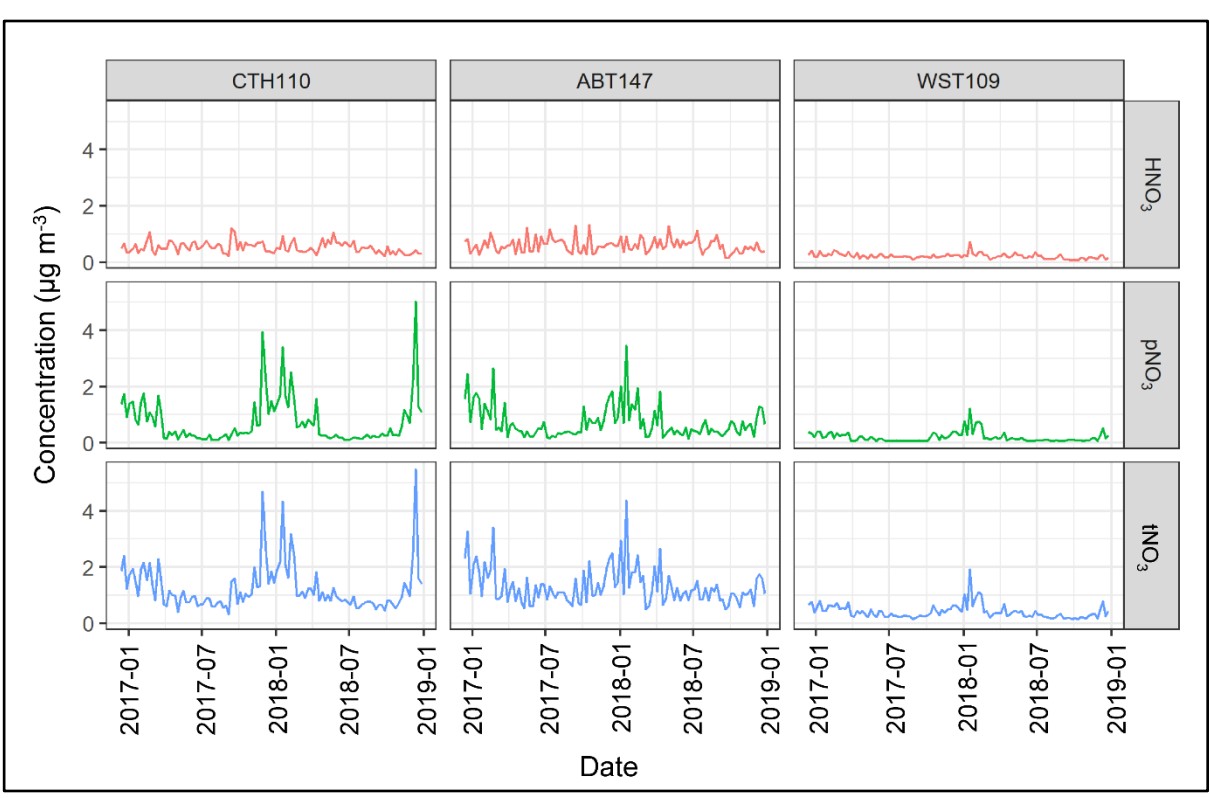

**Figure 3. Concentration data of nitric acid (HNO₃), particulate nitrate (pNO₃), and total nitrate (tNO₃ = HNO₃ + pNO₃) at the three CASTNET sites (Connecticut Hill, NY (CTH110) *N* =105, Abington, CT (ABT147) *N*=107, and Woodstock, NH (WST109), *N*=105) from December 2016 to 2018.**





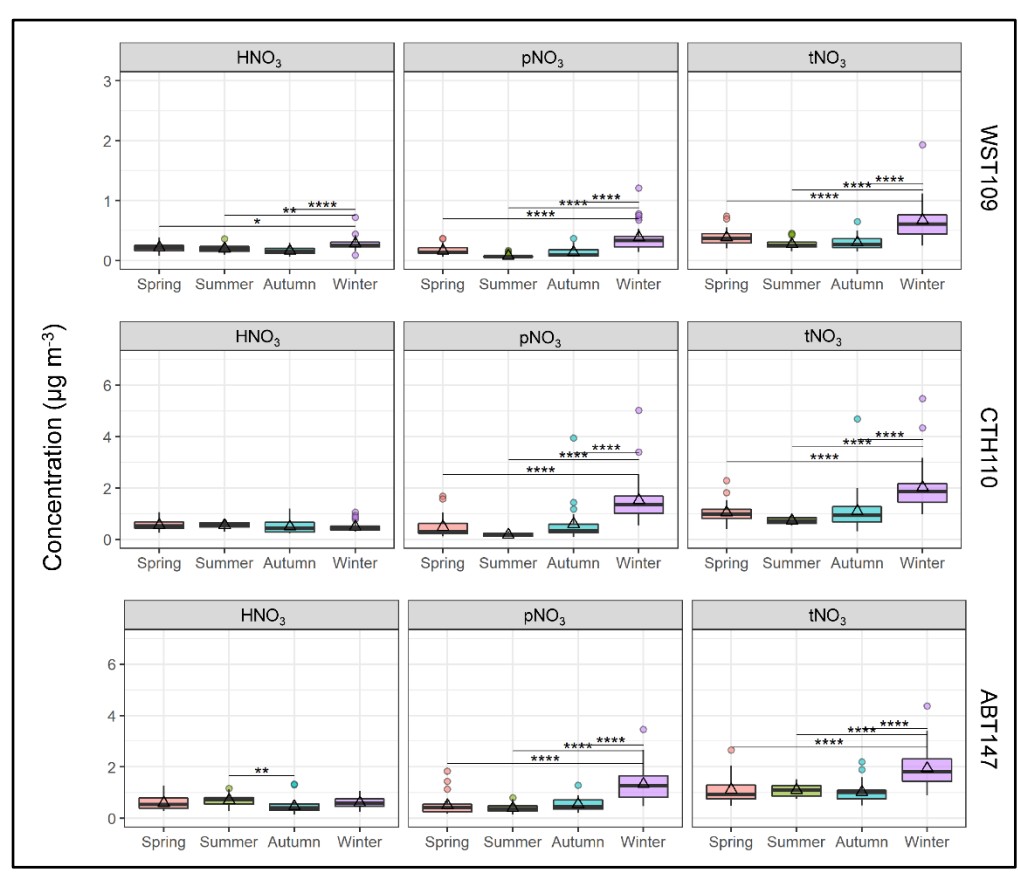

**Figure 4.** Box plot summary of seasonal concentrations of nitric acid (HNO₃), particulate nitrate (pNO₃), and total nitrate (tNO₃ = HNO₃ + pNO₃) at the considered CASTNET sites. The box plot summary indicates the distributions (lower extreme, lower quartile, median, upper quartile, and upper extreme) with the mean (open triangle) and outlier (data points) indicated. The p-values from ANOVA pairwise comparisons are indicated where *, **, ***, and **** indicate significant differences with p<0.01, p<0.001, p<0.0001, and p < 0.00001, respectively.

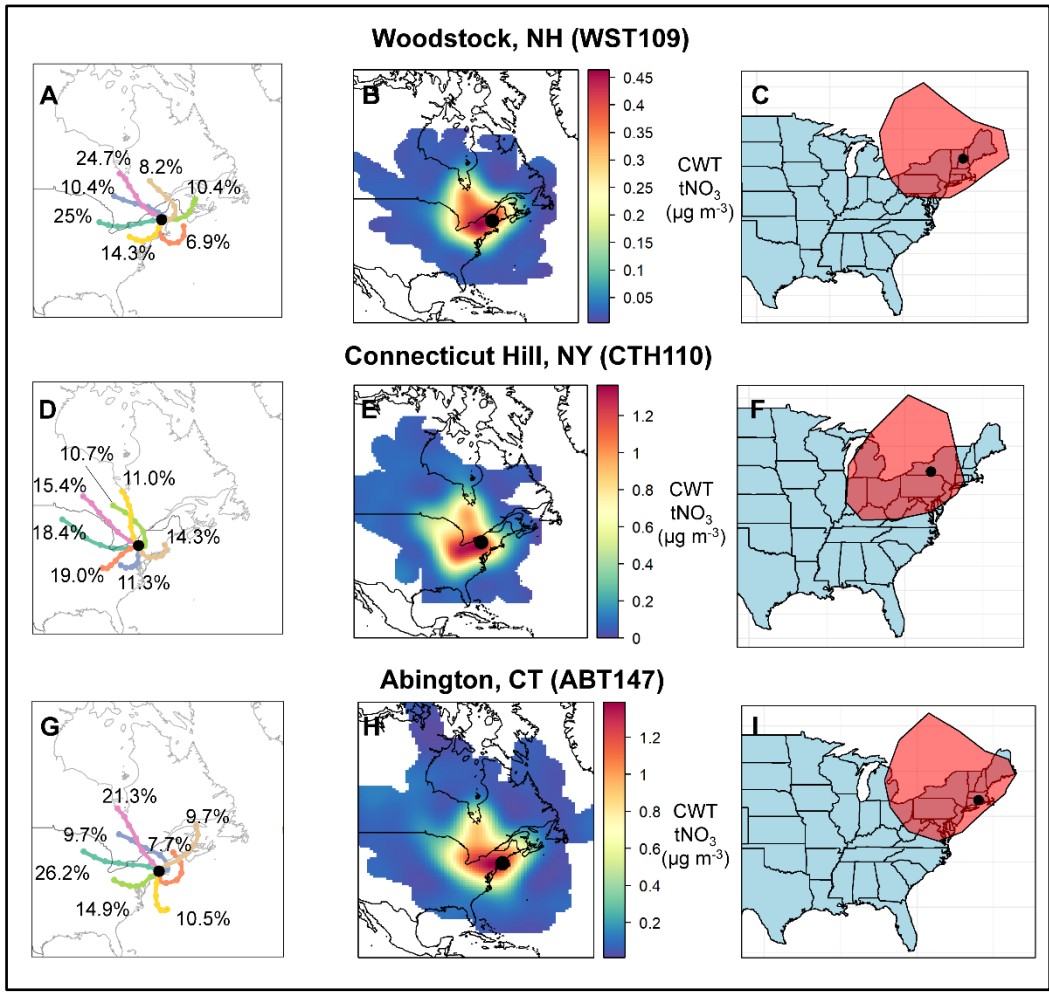

**Figure 5. Clustered air mass back trajectories (A, D, G), total nitrate (tNO$_3$ = HNO$_3$ + pNO$_3$) concentration weighted trajectories (B, E, H) and geospatial polygons (shown in red) representing the tNO$_3$ source contribution regions (C, F, I) at the CASTNET sites from December 2016 to 2018. The percentage contribution of each cluster to the total is also indicated.**




**Figure 6. Stable nitrogen isotope ($\delta(^{15}N)$) composition data of nitric acid (HNO$_3$), particulate nitrate (pNO$_3$), and total nitrate (tNO$_3$ = HNO$_3$ + pNO$_3$) at the three CASTNET sites (Connecticut Hill, NY (CTH110), Abington, CT (ABT147), and Woodstock, NH (WST109)) from December 2016 to December 2018.**







**Figure 7.** Box plot summary of seasonal $\delta^{15}N$ of nitric acid ($HNO_3$), particulate nitrate ($pNO_3$), and total nitrate ($tNO_3 = HNO_3 +$ $pNO_3$) at the considered CASTNET sites. The box plot summary indicates the distributions (lower extreme, lower quartile, median, upper quartile, and upper extreme) with the mean (open triangle) and outlier (data points) indicated. The p-values from ANOVA pairwise comparisons are indicated where *, **, ***, and **** indicate significant differences with $p<0.01$, $p<0.001$, $p<0.0001$, and $p < 0.00001$, respectively.




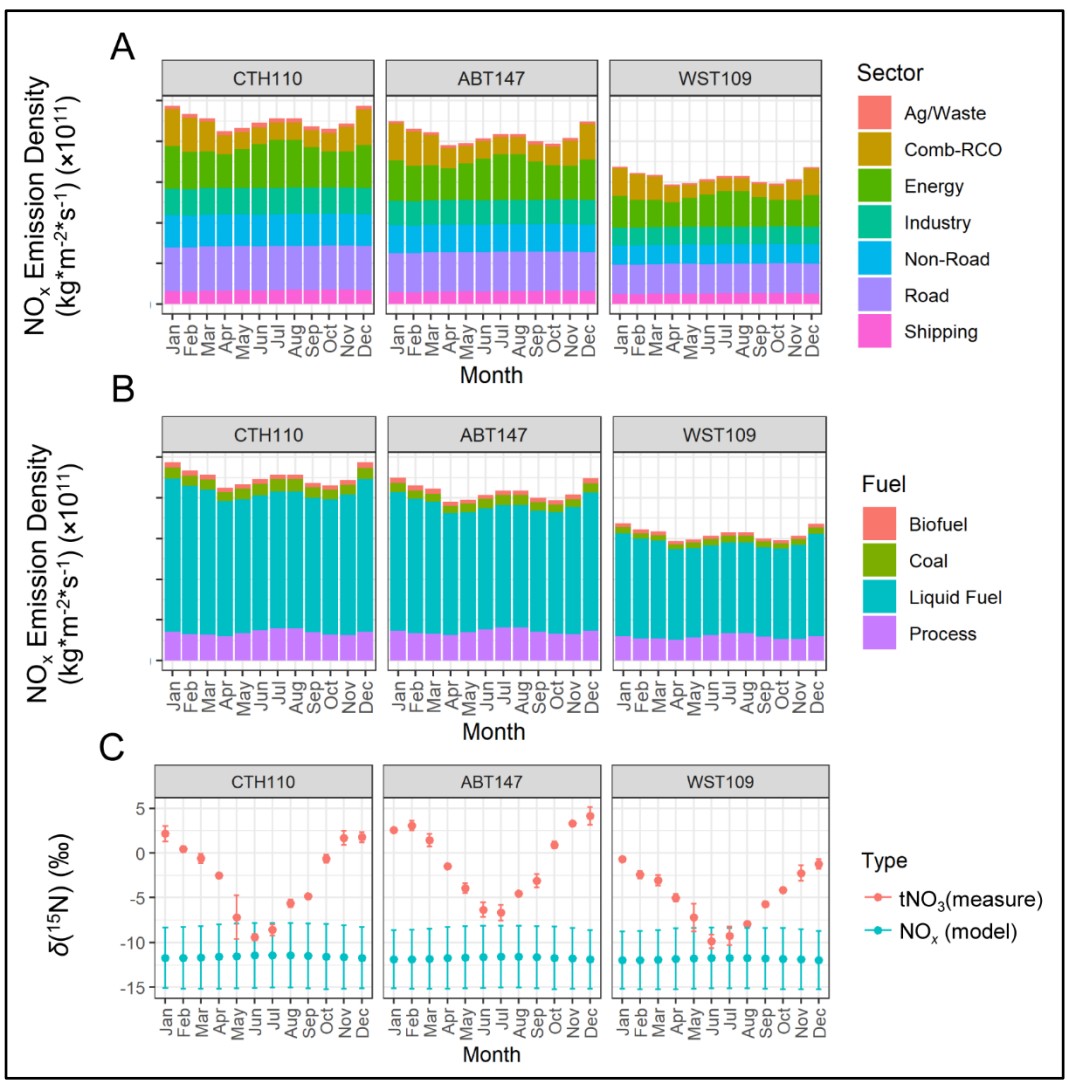

**Figure 8.** Estimated NO$_x$ emission density by sector (A) and fuel-type (B) for source regions contributing to the considered CASTNET sites, including Connecticut Hill, NY (CTH110), Abington, CT (ABT147), and Woodstock, NH (WST109). The monthly predicted $\delta(^{15}N, NO_x)$ from the emission estimates and the observed $\delta(^{15}N, tNO_3)$ are shown in C. The data points in C correspond to the mean, and the error bars correspond to the uncertainty, representing the propagated uncertainty for $\delta(^{15}N, NO_x)$ and the standard deviation for the $\delta(^{15}N, tNO_3)$ measurements.





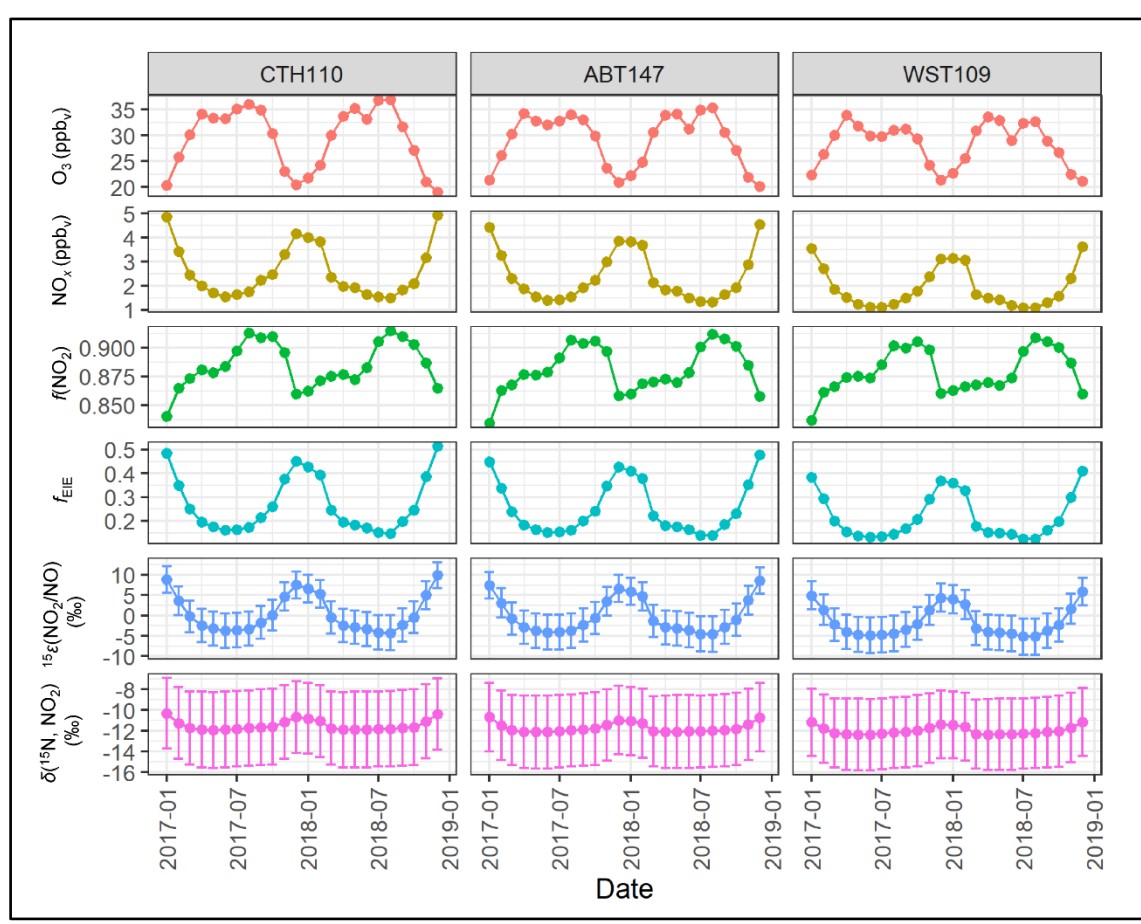

**Figure 9.** GEOS-Chem output of $O_3$, $NO_x$, and $f(NO_2)$ data and the calculated fraction of $NO_x$ at isotope equilibrium ($f_{EIE}$), the $NO_2/NO$ enrichment factor $^{15}\varepsilon(NO_2/NO)$, and $\delta(^{15}N, NO_2)$, at the considered CASTNET sites. The error bars in $^{15}\varepsilon(NO_2/NO)$ and $\delta(^{15}N, NO_2)$ correspond to the propagated uncertainty.





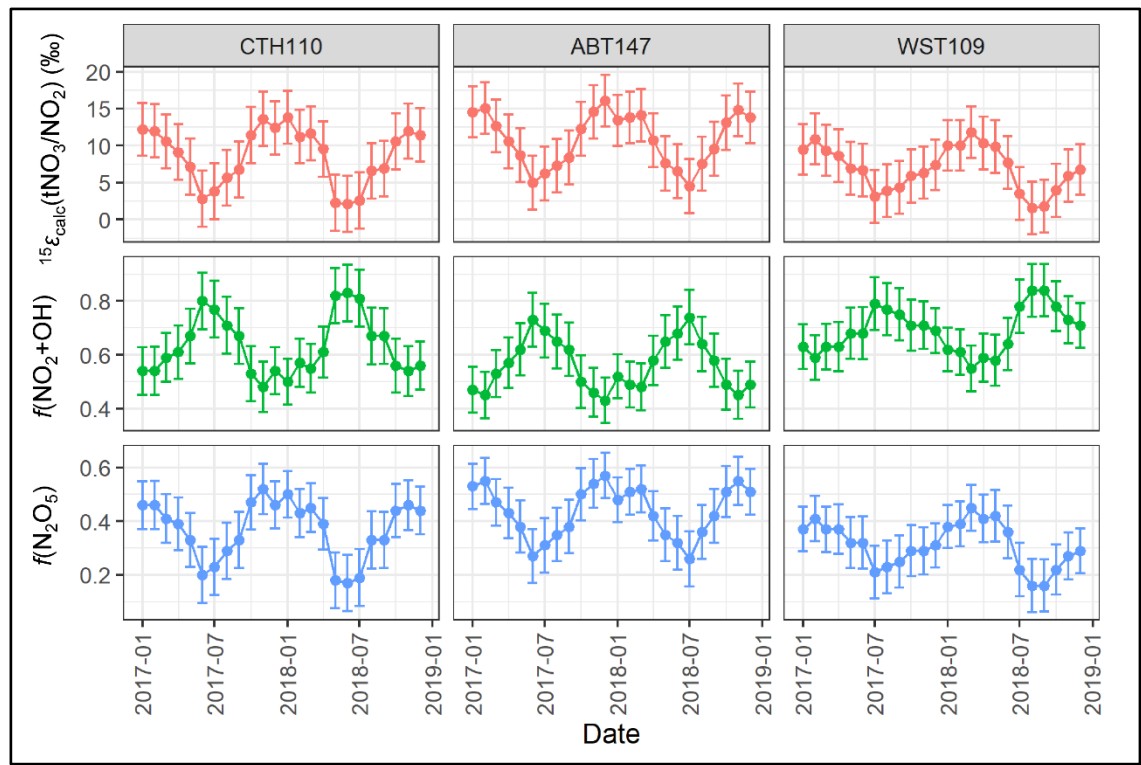

**Figure 10.** The calculated nitrogen enrichment factor associated with nitrate formation $^{15}\varepsilon(tNO_3/NO_2)$, and the estimated relative fraction of total atmosphere nitrate (tNO₃) formation via the N₂O₅ hydrolysis (R1) and NO₂ + OH (R2) pathways at the considered CASTNET sites. The error bars represent propagated uncertainty.