# Peer review of "Nitrate chemistry in the northeast US part I: nitrogen isotope seasonality tracks nitrate formation chemistry"

_Atmospheric Chemistry and Physics, 2022_

## Author Comment (AC1)

We appreciate both Reviewers' helpful, constructive, and insightful feedback, which has helped improve our manuscript. Overall, both reviewers commented on the significance and interest of the presented work but recommended further expansion of the discussion section and sampling methodology. To this end, we have added further discussion on the drivers of the observed spatiotemporal differences in atmospheric nitrate concentrations and $\delta(^{15}N)$. Additionally, we expanded our discussion of the drivers of the seasonal variation in atmospheric nitrate production and their implications for atmospheric chemistry. We have also provided further detail on the US EPA CASTNET sampling protocol. In the revised manuscript, we have also majorly improved the quality and presentation of the Figures. Overall, these changes have led to the improvement of the presented manuscript. A point-by-point response to all reviewer comments is provided below.

**Reviewer #1:**

**Overview:** The authors present new isotopic data and model calculations for atmospheric nitrate phases in the northeastern USA. The results reveal seasonal isotopic patterns in the measured data, which cannot be explained by source variability alone. Instead, the authors propose that the isotopic composition of atmospheric nitrate is significantly impacted by secondary reactions within the atmosphere and that the temporal variability can be explained by changes in the formation pathway.

The manuscript is well written and underpinned by a strong dataset. I'm not an atmospheric chemist and cannot evaluate the model calculations, but as an isotope geochemist I found the paper interesting to read and overall compelling.

My main comment is to better discuss the importance of varying nitrate formation pathways. What causes them to change seasonally, and what can that tell us about the broader environment (either anthropogenic or natural processes)?

Apart from that, I only have a few minor points for clarification (to the non-specialist):

**Response:** We thank the reviewer for their helpful and constructive feedback. We have addressed the raised points and revised the manuscript according to these suggestions. These changes have improved the clarity of the revised manuscript. Below is a point-by-point response to the raised comments.

**Comment:** l. 60-61: This sentence needs to be simplified. Maybe write "Accounting for these isotope effects is important when using for $\delta(15N)$ as quantitative tracker…"

**Response**: Thank you for the suggestion. We have revised this sentence to the following, "Accounting for these isotope effects is important for $\delta(^{15}N)$ to be used as a quantitative tracker of precursor emission sources". These changes have been made on Lines 61-62 in the revised manuscript.

**Comment**: l. 85-86: This sampling description is lacking lots of details. Which containers were used? How were the filters applied? How much sample was collected? Please expand this paragraph and provide appropriate references.

**Response**:  Thank you for pointing out the lack of methodological details. In the revised manuscript, we have added the following to describe the CASTNET sampling methodology further: "The CASTNET sampling protocols have been previously described (Baumgardner et al., 2002). The atmospheric samples consist of week-long integrated collections using a three-stage

filter pack. The filter pack contains three types of filters in sequence: (1) a Teflon filter (Whatman membrane filter, 47 mm diameter, 1.0 μm pore size) for particulate collection, including $pNO_3$; (2) a Nylon filter (before January 2018: Pall Corporation Nylasorb, 47 mm diameter, 1.0 μm pore size; after January 2018: One Measurement Technology Laboratories, 47 mm diameter, 1.0 μm pore size) for acidic gas collections, including $HNO_3$; and (3) two potassium carbonate ($K_2CO_3$) impregnated cellulose filters (Whatman 41 Ashless Circle filter) for $SO_2$ collection. The filter pack sampling system is characterized as "open faced", because a size-selective inlet is not used. The filter packs are prepared and shipped to the field weekly. The filter packs are exchanged at the sampling sites every Tuesday and shipped to the analytical chemistry laboratory for analysis. Blank filter packs are prepared quarterly to evaluate contamination. The filter pack samples are collected at 10 m, and the filter pack flow rate is maintained at 1.50 L min at standard conditions. The filters were extracted and analyzed for concentrations following standardized protocols at the Wood Gainesville, FL, US laboratory. Briefly, the filters were extracted using 25 mL of MQ water, and the Teflon and Nylon filter extracts were measured using a micro membrane-suppressed ion chromatography to determine $NO_3^-(aq)$ concentrations, which were utilized to calculate the concentration of $pNO_3$ and $HNO_3$ in the air ($\mu g\ m^3$) based on the volume of collected air. Following this analysis, the samples were stored in a laboratory at room temperature for up to two years until shipment to Brown University". These additions were made on Lines 89-104 in the revised manuscript.

**Comment**: l. 93: Perhaps also state that the data agree to within XX%. A 1:1 relationship could also arise if there were a significant but consistent offset between the datasets.
**Response**: Thank you for raising this point. We have provided additional information about the re-measured $NO_3^-(aq)$ concentrations relative to the values reported by CASTNET. The following sentence was added on Lines 113-115 in the revised manuscript, "Additionally, the mean absolute difference and the mean percent difference between the re-measured and reported $NO_3^-$ (aq) concentrations were ($0.31 \pm 0.36$ μM; $\bar{x} \pm \sigma$) and ($10.4 \pm 13.3$ %), respectively ($n = 632$)."

**Comment:** ll. 94-95: Does this mean that samples from four weeks were mixed together into one container? Please state this more clearly.
**Response**: Thank you for pointing out that this line was unclear. We have revised the sentence to the following on Lines 115-116 in the revised manuscript, "Equal volumes of four weekly-collected filter extracts were combined into approximately monthly aggregates to provide sub-seasonal resolution of nitrogen isotope analysis for $HNO_3$ and $pNO_3$." Further, we highlighted the reason for combining samples into monthly aggregates, which was due to sample mass requirements for isotope analysis on Lines 116-117 in the revised manuscript, "Samples were combined into month aggregates to meet the typical mass requirements for isotope analysis, requiring 20 nmol for $\delta(^{15}N)$ and $\delta(^{18}O)$ and 50 nmol for $\Delta(^{17}O)$ quantification."

**Comment:** ll. 104-105: Name the model of the mass spectrometer. Was the same instrument used to measure D17O?
**Response:** Thank you for pointing out this oversight. In the revised manuscript, we have indicated the model of the mass spectrometer (Thermo Scientific Delta V) on Line 127, "..(CF-IRMS; Thermo Scientific Delta V)…". Further, we have identified the instrument used to analyze $\Delta(^{17}O)$, "The generated $O_2$ was introduced to a CF-IRMS (Thermo Scientific Delta V) and measured at

*m/z* 32, 33, and 34 for $\Delta(^{17}O)$ (defined as: $\Delta(^{17}O) = \delta(^{17}O) - 0.52 \times \delta(^{18}O))$ determination (Kaiser et al., 2007).". This change was made on Line 136 in the revised manuscript.

**Comment:** l. 185: Provide some quantitative comparison here for how high NOx levels used to be in the past. Otherwise, the reader is left wondering.
**Response:** Thank you for pointing this out. In the revised manuscript, we have quantified the change in $NO_x$ emissions in the US based on a recent study (Miyazaki et al., 2017) utilizing top-down satellite observations. This change was made in Lines 219-221 in the revised manuscript, "Thus, even as $NO_x$ emissions have dramatically decreased in the US by 38 % from 2005-2014 as evidenced from top-down global surface $NO_x$ observations (Miyazaki et al., 2017), the $HNO_3$ and $pNO_3$ seasonal trends in the northeast US have been retained."

**Comment:** l. 228: Better rephrase to "This increase is likely due to a significant heating demand during this period". Please also explain why other reasons can be ruled out, leaving heating (i.e., coal combustion?) as the most likely explanation. NOTE: Later in the discussion it is argued that the observed seasonal variability is caused by secondary fractionation effects rather than source variability. Hence this sentence here should be rewritten. Otherwise, it is very confusing.
**Response:** Thank you for pointing out that this section was confusing. The referenced sentence describes the $NO_x$ emission amounts rather than the modeled isotope values of $NO_x$ emissions. These emission amounts represent the output of the Community Emissions Data System ((McDuffie et al., 2020). The original Figure 8A and 8B displayed the estimated $NO_x$ emission densities, such that the higher combustion activity during winter and summer is not assumed but is based on the $NO_x$ emission model (McDuffie et al., 2020). To further clarify this section, we have attempted to make it more obvious that this section is based on the model output of the Community Emission Data System ("To test this hypothesis on the current dataset, the monthly predicted $NO_x$ emission densities speciated by sector and fuel-specific sources based on the Community Emissions Data System"). Further, we indicated the units of the emission densities $(kg\ m^{-2}\ s^{-1})$. Additionally, we revised the original Figure 8 (now Figure 5) to only include the $NO_x$ emission density estimates and not include the $\delta(^{15}N)$ data, which is now presented in a separate figure (Figure 6). These changes were made on Lines 258-261 in the revised manuscript.

**Comment:** l. 229: As above, change to "… possibly due to increased emissions related to electricity generation for cooling". Explain why other reasons can be ruled out. NOTE (same as above): Later in the discussion it is argued that the observed seasonal variability is caused by secondary fractionation effects rather than source variability. Hence this sentence here should be rewritten. Otherwise, it is very confusing.
**Response:** Please see our response to the previous comment as it was directly related to this comment. We would like to point out that we are not speculating about the reasons the emission amounts are increasing but indicating, based on the model, the seasonal changes in the predicted $NO_x$ emissions. We clarify that we are referring to $NO_x$ emission amounts by defining the units $(kg\ m^{-2}\ s^{-1})$ and referring to the emission amounts in the revised Figure 5.

**Comment:** ll. 231-238: How could all these endmembers be calculated so accurately? The introduction mentions isotope data for only soils, liquid fuel combustion, vehiclces and coal combustion. How were all these other sources listed here isolated from mixed isotopic signals? Please explain how this was done.

**Response:** Thank you for raising this point. This section refers to the predicted $NO_x$ emissions of the Community Emissions Data System (McDuffie et al., 2020). We have summarized the predicted sector and fuel-based $NO_x$ emissions for our study region. We have clarified in the revised manuscript that this section is based on predictions by the Community Emissions Data System and refer to Figure 5, which was based solely on the emission model output. These revisions were made on Lines 267-271, "Across the three sites, the Community Emissions Data System predicts that there were similar annual contributing $NO_x$ emission sectors for the identified source regions contributing $tNO_3$ to the study sites (CTH110, ABT147, WST109) that included energy (21.9 %, 22.5 %, 23.5 %), industry (14.4 %, 14.6 %, 14.1 %), non-road transport (17.3 %, 16.2 %, 15.0 %), combustion-residential, commercial, other (12.8 %, 14.2 %, 14.3 %), road (23.9 %, 23.2 %, 23.3 %), shipping (7.5 %, 7.5 %, 8.5 %), and agricultural/waste (2.1 %, 1.7 %, 1.5 %) (Figure 5). Additionally, there was similar annual $NO_x$ emission density contributing fuel-types across sites, including Biofuel (2.6 %, 2.7 %, 2.7 %), Coal (5.8 %, 5.2 %, 4.8 %), Liquid-fuel (76.4 %, 75.0 %, 73.9 %), and Process-based emissions (15.3 %, 17.2 %, 18.7 %) for the identified source regions contributing to $tNO_3$ at CTH110, ABT147, and WST109, respectively (Figure 5)".

**Comment:** l. 276: parenthesis missing in the equation
**Response:** Thank you for catching this error. We have made the correction on Line 316 in the revised manuscript.

**Comment:** l. 285: change fractionation to reaction
**Response:** Thank you for this suggestion. However, in this sentence, we are referring to the impact on $\delta(^{15}N)$, such that I think fractionation is the correct term.

**Comment:** l. 289: How low is low? Provide a quantitative threshold, so that one can compare this to the data.
**Response:** Thank you for pointing out this uncertainty. We have revised the sentence to the following on Lines 328-329 in the revised manuscript, "In contrast to the EIE, the LCIE dominates $NO_x$ $\delta(^{15}N)$ fractionation during conditions of higher $O_3$ concentrations relative to $NO_x$ concentrations (Li et al., 2020)."

**Comment:** ll. 331-332: How are the pathways changing? Please expand briefly.
**Response:** Thank you for pointing out this ambiguous wording. We have modified this section to more clearly indicate that the nitrate formation pathway is changing from $NO_2$ + OH and $N_2O_5$ heterogeneous chemistry. We revised this sentence to the following, "The shift in the seasonal $^{15}\varepsilon_{calc}(tNO_3/NO_2)$ was likely attributed to a change in the dominant nitrate formation pathway from $NO_2$ oxidation via hydroxyl radical during the summer to increased $N_2O_5$ hydrolysis during the winter, as previously suggested (Li et al., 2021) and in our companion study (acp-2022-622)." This change was made on Lines 374-376 in the revised manuscript.

**Comment:** l. 362: Again, remind the reader here how large this reduction was, in percent.
**Response:** Thank you for pointing this out. We have quantified the reduction of $NO_x$ emissions from 2005-2014 in the US. This sentence has been revised to the following in the revised manuscript, "These findings were consistent with a previous study of atmospheric nitrate from CASTNET sites collected in the early 2000s, indicating that even after dramatic reductions in $NO_x$ emissions in the US over the past decade (e.g., a decrease of 38 % from 2005-2014; Miyazaki et

al., 2017), atmospheric nitrate spatiotemporal trends have been retained." This change was made on Line 415-418.

**Comment:** ll. 372-374: What is the importance of knowing the nitrate formation pathway? Can knowledge over the pathway help understand air quality or other parameters? Please expand.
**Response:** Thank you for raising this point. We added the following to this section to raise the importance of tracking nitrate formation, "Tracking the formation pathways of nitrate is important for evaluating atmospheric chemistry model representation of oxidation chemistry. For example, uncertainties in the rate of $NO_x$ oxidation to nitrate have been shown to represent a significant source of uncertainty for the formation of major tropospheric oxidants (i.e., ozone ($O_3$) and the hydroxyl radical (OH)) that has important implications for our understanding of atmospheric lifetimes of many trace gases, including greenhouse gases". This change was made on Line 430-434.

**Comment:** Figure 3: Is there any significance to the nitrate speciation between HNO3 and particulate? Does that relate back to the nitrate formation pathway as well? This would be helpful to discuss in the manuscript.
**Response:** Thank you for this comment. The significance between $HNO_3$ and $pNO_3$ is that they have different atmospheric lifetimes (i.e., $HNO_3$ can undergo dry deposition at a much faster rate) and that $pNO_3$ contributes to particulate matter pollution. We have clarified this point in the revised manuscript on Lines 202-205, "The speciation of $tNO_3$ concentration is important to evaluate due to $HNO_3$ and $pNO_3$ different atmospheric lifetimes driven by deposition rates (Benedict et al., 2013). Due to a higher dry deposition rate, $HNO_3$ has a shorter atmospheric lifetime of a few days (i.e., 1-3 days) relative to $pNO_3$, which has a lifetime of several days (i.e., 5 to 15 days)."
For evaluating nitrate formation pathways based on $\delta(^{15}N)$, we did not separate $HNO_3$ and $pNO_3$. This is because of the potential $\delta(^{15}N)$ phase fractionation between $HNO_3$ and $pNO_3$, which was indicated in the original manuscript (Lines 369-370 in the revised manuscript).

We note that differences in formation pathways for $HNO_3$ and $pNO_3$ were evaluated in our companion paper (acp-2022-622). We clarified this point in Lines 367-368 in the revised manuscript, "We acknowledge there could be potential differences in formation pathways for the speciated phases of atmospheric nitrate (i.e., $HNO_3$ and $pNO_3$). However, we evaluated nitrate formation from the mass-weighted $\delta(^{15}N, tNO_3)$ to remove the potential $\delta(^{15}N)$ phase-fractionation between $HNO_3$ and $pNO_3$, which complicates evaluating the potential phase-dependent formation pathway."

**Comment:** Figure 4: Make the y-axes shorter in all plots, so that the data are more spread out.
**Response:** Thank you for pointing this out. In the revised manuscript, we removed the original Figure 4, since we felt its information was redundant with the other figures and tables.

**Reviewer #2:**

In this manuscript, the authors examine atmospheric nitrate concentrations and $d^{15}N$ values at three sites in the northeastern United States over two years. They find clear seasonal cycles in total $NO_3^-$ concentration and in $d^{15}N$ values. Modelling suggests that these cycles cannot be explained by seasonally changing $NO_x$ sources or by isotopic fractionation during $NO_x$ cycling. Rather, isotopic fractionation during the formation of nitrate provides the best match to observed $d^{15}N$ values and thus seasonally changing nitrogen formation pathways appear to drive $d^{15}N$ variability over a year. This is in contrast to previous studies which more prominently focused on $NO_x$ emissions as the key $d^{15}N$ driver.

Overall, I found the manuscript to be well-written and generally clear. The narrative is straightforward and generally well-balanced between providing enough information to follow the methodology and authors' thoughts while not getting too bogged down in technical and modelling details. The work appears to have been performed well with a comprehensive set of analyses and models to investigate this field data. The figures are generally good, although Figure 1 needs substantial changes to meet the quality shown throughout the rest of the manuscript.
My comments are generally minor. My largest concern is that while the paper states that it is examining spatiotemporal variability, there is actually very little discussion of the spatial variability between sites. The WST site, in particular, has much lower $NO_3^-$ concentrations than the other two sites, and smaller but consistent differences in $d^{15}N$ values are also mentioned across the three sites. Are there geographical or environmental differences between these sites that could explain these observations? Can these differences help you understand or interpret the model predictions better? Also, be careful describing things as spatiotemporal variability if you actually just discussing a broad temporal variability observed in a similar manner across the three sites.

Additionally, while the authors identify that the $NO_3$ formation pathways are the most likely drivers of the temporal variability, the final discussion could use a bit more elaboration (for those with less expertise in atmospheric chemistry and modelling) on why these formation pathways differ seasonally. Also, it could be useful if the authors spoke to whether these seasonal pathways are expected to have been affected by air pollution and other atmospheric chemistry changes over the past several decades (i.e., would you expect these to have largely had the same effect on $NO_3^-$ isotopes in the pre-industrial or during the peak of the $NO_x$ pollution as now?). This need not be an intense discussion, but rather adding some context for readers to better understand how applicable your findings could be and get a better idea of the importance.
**Response:** We thank the reviewer for their helpful and constructive feedback. We have addressed the raised points and revised the manuscript according to these suggestions. These changes have improved the clarity of the revised manuscript. Below is the point-by-point response to the raised comments. Additionally, we want to acknowledge that we have addressed the reviewer's concern regarding the spatiotemporal variations in atmospheric nitrate. In the revised manuscript, we have provided a more detailed discussion regarding the spatial variations observed across the CASTNET sites in our study. These changes are summarized in our responses to the specific comments raised by Reviewer #2.

In addition, we have provided further context regarding the seasonal variation in atmospheric nitrate formation. We have added the following lines in the discussion section, "This seasonality

in atmospheric nitrate formation is driven by photochemistry and temperature. The OH radical is formed via photolysis, so its abundance is greater during the summer, leading to a relative increase in the proportion of atmospheric nitrate formed via $NO_2 + OH$ homogenous (gas phase) reactions. During the nighttime, higher order nitrogen oxides form, and new pathways of atmospheric nitrate production become important. Under these conditions, $NO_2$ is oxidized by $O_3$, forming the nitrate ($NO_3$) radical, which exists at thermal equilibrium with $NO_2$ and $N_2O_5$, which can subsequently hydrolyze on wetted aerosol surfaces leading to atmospheric nitrate production. $N_2O_5$ is photolabile and thermally unstable, so $N_2O_5$ heterogeneous reactions on aerosol surfaces are typically most prevalent during the winter (Alexander et al. 2020)." These updates are reflected in Lines 401-408 in the revised manuscript.

Further, we highlighted the importance of tracking atmospheric nitrate formation pathways in the conclusion section, "Tracking the formation pathways of nitrate is important for evaluating atmospheric chemistry model representation of oxidation chemistry. For example, uncertainties in the rate of $NO_x$ oxidation to nitrate have been shown to represent a significant source of uncertainty for the formation of major tropospheric oxidants (i.e., ozone ($O_3$) and the hydroxyl radical (OH)) that has important implications for our understanding of atmospheric lifetimes of many trace gases, including greenhouse gases." These additions were made on Lines 430-434 in the revised manuscript.

**Specific comments**

**Comment:** Notation for $d^{15}N$ is non-standard and should be simply $d^{15}N$. For specific chemical species, notation could either be $d^{15}N(NO_x)$ or $d^{15}N_{NOx}$.
**Response:** We appreciate the feedback; however, this notation we have used follows IUPAC recommendations that for any quantity symbol, including isotope deltas, to enclose labels in parentheses. Additionally, this was the recommended notation by the editor. Thus, we did not make these suggested notation changes.

**Comment:** 21: I know this is just the abstract, but "Instead, the spatiotemporal trends were driven by $\delta(15N)$ fractionation associated with formation" is pretty vague for being one of the primary findings of your paper. Could you give a little more specific information on what part of the formation process or reaction is driving this fractionation?
**Response:** Thank you for this comment, and we agree that the wording was vague in the original manuscript. To improve the clarity of this statement, we have modified the sentence as follows, "Instead, the seasonal and spatial differences in the observed $\delta(^{15}N)$ of atmospheric nitrate were driven by nitrate formation pathways (i.e., homogenous reactions of $NO_2$ oxidation via hydroxyl radical or heterogeneous reactions of dinitrogen pentoxide on wetted aerosol surfaces) and their associated $\delta(^{15}N)$ fractionation.". This change was made on Lines 21-23 in the revised manuscript.

**Comment:** 80: What are the elevations for these three sites? I see that they are located away from cities and point source pollution, but are they in primarily agricultural or natural settings? Do the three sites differ by surrounding land use or other environmental factors other than geographic coordinate?

**Response:** Thank you for raising this point. We have provided more details regarding the sample sites in the revised manuscript, including elevation and characteristics. Further, we have provided a link to the CASTNET sample sites. These changes were made on Lines 80-84 in the revised manuscript, "Filter samples from December 2016 to 2018 were obtained from the US EPA CASTNET program for several sites in the northeastern US, including (from West to East) Connecticut Hill, NY (CTH110; 42.40° N, -76.65° W; Elevation = 511 m), Abington, CT (ABT147; 41.84° N, -72.01° W; Elevation = 202 m) and Woodstock, NH (WST109; 43.94° N, -71.70° W; Elevation = 255 m) (Figure 1). The CASTNET sites were characterized by their primary land use as forest for CTH110, urban/agricultural for ABT147, and forest for WST109, respectively (CASTNET Site Locations, 2023)."

**Comment:** 107: This is a fairly small range of $d^{15}$N values for your standards, and many of your samples have $d^{15}$N values outside this calibration standard range. Can you speak to the quality of the corrections outside this range? Is there solid reason or evidence to assume that an extrapolated correction is still accurate and precise?

**Response:** Thank you for pointing this out. We acknowledge that the range of the $\delta(^{15}$N) values of the standards (from -1.8 to 4.7 ‰) was relatively narrow. However, we point out that the range of our measurements was near the values of the standards (samples ranged from -10.6 to 5.8 ‰ and averaged -1.7 ± 3.7 ‰; $n$=158). We have made the following additions to the revised manuscript to point this out in Lines 131-134, "We acknowledge that the $\delta(^{15}$N) range of the nitrate reference material is relatively narrow; however, the range of our calibrated unknowns was quite near these values (calibrated unknowns ranged from -10.6 to 5.8 ‰ and averaged -1.7 ± 3.7 ‰; n=158). Thus, while some of the unknowns will have a calibrated $\delta(^{15}$N) extrapolated from the reference materials, we do not anticipate this to impact our measurement accurary and precision or the interpretation of the results".

**Comment:** 175: There's no discussion of why there is such a dramatic difference in concentrations between WST and the other two sites. This would seemingly indicate that either the $NO_3$ supplies and/or flux are very different for this site. Would this affect your ability to make region-wide conclusions? To that end, how much spatial variance in $NO_3$ concentration and $d^{15}$N are you expecting to see across the region? Should they all be fairly similar, or might you expect substantial local variations?

**Response:** Thank you for raising this point. We have drawn attention to the significant spatial differences in nitrate concentrations in the revised manuscript. The following additions were made on Lines 206-210, "Lower nitrate concentrations at the Woodstock, NH site compared to the other site likely reflects the different amounts of $NO_x$ emissions and thus the amount of nitrate impacting the study sites. For example, the Woodstock, NH site is relatively remote compared to the urban/agricultural characterization of Abington, CT and Connecticut Hill, NY, which is directly downwind of the highly industrialized Ohio River Valley and other midwestern cities."

Further, we have added discussions of our air mass back trajectory analysis, which indicated the role of transport patterns and the positioning of the sites on the impact of nitrate concentrations. This addition was made on Lines 229-230 in the revised manuscript, " .. due to the location of the sites, which likely impacts the observation nitrate concentration trends observed at the sites".

Lastly, we evaluated $NO_x$ emission density for the source regions contributing nitrate to the study sites. Our analysis indicated that the $NO_x$ emission density was lower for Woodstock, NH, than the other sites. In the revised manuscript, we provided a more direct link between this emission modeling and the concentration results on Lines 265-267, "The absolute $NO_x$ emission densities were higher for CTH110 and ABT147 compared to WST109 (Figure 8A-B), which may explain the observed nitrate concentration trends with the lowest concentrations observed at WST109 (Figure 1)."

While there was a spatial difference in $NO_x$ emission densities and the nitrate concentrations at the considered study sites, the modeled $\delta(^{15}N, NO_x)$ were nearly identical for all sites. This is because the relative fuel breakdown of $NO_x$ emissions contributing to each study site was nearly identical. Thus, the significant differences observed in $\delta(^{15}N)$ of nitrate across our sites could not be related to precursor sources. Slight differences in oxidation chemistry drive this difference. To make these points clear, we have made a few additions in the revised manuscript. On Lines 278-282 we added, "We note that while there were significant differences in modeled $NO_x$ emission densities and observed nitrate concentrations at the study site, the relative contributions of $NO_x$ emissions contributing to the study sites were nearly identical leading to similar modeled $\delta(^{15}N, NO_x)$ values. Thus, $NO_x$ emissions were not the main contributor to the observed spatial differences in $\delta(^{15}N,$ $HNO_3, pNO_3, tNO_3)$". On Lines 396-398 we added, "This calculation suggests that the observed temporal $\delta(^{15}N)$ differences at the considered sites were driven by slight differences in nitrate formation and oxidation chemistry."

**Comment:** 200: Again, there's no discussion of what might be causing these spatial differences in the $d^{15}N$ values between sites.
**Response:** Thank you for this comment. We have addressed this lack of discussion in the original manuscript in our response to the previous comment. To reiterate, the significant spatial differences observed in $\delta(^{15}N, HNO_3, pNO_3, tNO_3)$ were not driven by differences in relative contributions of $NO_x$ emissions (by sector and fuel type). This was because the relative $NO_x$ emission contributions were nearly identical for the study sites leading to nearly identical modeled $\delta(^{15}N, NO_x)$. The significant spatial difference is thus most likely driven by oxidation chemistry. While our mass-balance calculation suggests that nitrate formation pathways had similar temporal trends across the considered sites (i.e., more nitrate formed via $N_2O_5$ heterogeneous reactions during winter and more nitrate formed via $NO_2 + OH$ reactions during summer), the absolute contributions leading to nitrate formation were different between sites, explaining the spatial $\delta(^{15}N,$ $HNO_3, pNO3, tNO_3)$ patterns.

**Comment:** 205: This close match to theoretical values is nice to see in the field data!
**Response:** Very cool indeed!

**Comment:** 255: Positively or negatively correlated?
**Response:** Thank you for pointing this out. In the revised manuscript, we have indicated the correlations were positively correlated. This change was made on Line 295 in the revised manuscript.

**Comment:** Figure 1. This map needs substantial changes or wholesale remaking, as it is not up to the quality of the rest of the manuscript. There are no indications for the symbology and coloring

of the map. There is no scale for the map, nor any indication of the particular projection of the map (it is probably Web Mercator based on Google Maps, but this is generally a poor projection choice for scientific use). Likewise, there are no geographic coordinates nor an inset map indicated to give geographical context to the selected area of the map. These introduction map figures give good opportunity to provide supplemental environmental and geographic information about the region, such as elevation or land cover, and as such you should consider having this figure give the reader more context and information than simply the three locations on a transportation map.
**Response:** Thank you for this comment. In the revised manuscript, we have significantly improved Figure 1. We have added coordinates and labels and provided a world map insert of our study location. To provide a further context of the study site locations and nitrate concentrations, we added the concentration data (from Figure 3) into the figure. Overall, Figure 1 is much improved in the revised manuscript.

**Comment:** Figure 5: The maps in C, F, and I should also have Canada and Mexico present, particularly since the source regions extend into Canada.
**Response:** Thank you for pointing this out. We have expanded the map of panels C, F, and I in the revised manuscript. The updated figure is now Figure 3 in the revised manuscript.

**Technical corrections**

**Comment:** This may be caught in later proofing, but I believe the editorial style for Copernicus is one space after periods, and here throughout it appears to be 2 spaces (and in some cases, maybe 3 spaces).
**Response:** Thank you for pointing this out. We have revised the sentence spacing to 1 space.

**Comment:** 12: Comma after Here
**Response:** Thank you for catching this typo. We have made this correction on Line 12 in the revised manuscript.

**Comment:** 30: Consider writing this as "total atmospheric nitrate" so that it is more clear why the abbreviation has a "t" in it.
**Response:** Thank you for this comment. We have updated our terminology as "total atmospheric nitrate" on Line 31 in the revised manuscript.

**Comment:** 103: P. aureofaciens should be italicized. And the genus should probably be fully spelled out since it is not spelled out elsewhere. It may also be worth noting that this is a specific modified strain of the bacteria to lack N2O reductase, and not simply the wild-type.
**Response:** Thank you for pointing this out. In the revised manuscript we have provided the full name and italicized *Psudomonas aureofaciens*. Additionally, we have added that this strain lacks the $N_2O$ reductase enzyme. These revisions were made on Lines 125-126.

**Comment:** 156: I think Copernicus publications usually have these model Zenodo DOI citations included in the literature citations.
**Response:** Thank you for pointing this out. We have revised our citation of GEOS-Chem in the revised manuscript in Lines 181-185, "The GEOS-Chem global model of atmospheric chemistry (www.geos-chem.org) was utilized to predict $NO_x$ and $O_3$ concentrations in the regions of the

various CASTNET sites (Bey et al., 2001; Walker et al., 2012, 2019). The model was utilized to account for $\delta(^{15}N)$ isotope fractionation that occurs during chemical reactions. We use version 13.2.1 (http://wiki.seas.harvard.edu/geos-chem/index.php/GEOS-Chem_13.2.1) of the model driven by GEOS5-FP assimilated meteorology from the NASA Global Modeling and Assimilation Office (GMAO)".

**Comment:** 204: Parenthesis aren't needed when simply stating a mean value difference.
**Response:** Thank you for the suggestion; however, this notation was recommended by the editor, such that no change was made.

**Comment:** Table 2: Standard deviations are more easily and typically shown as ± after the mean (i.e., Mean±SD, 0.526±0.200)
**Response:** Thank you for this suggestion. We have changed the notation in Table 2 of the revised manuscript as suggested.

**Comment:** Figure 4: Is there a true need to distinguish between $p < 0.001$, $p < 0.0001$, and $p < 0.00001$? Are these scientifically significant differences?
**Response**: Thank you for this comment. We removed Figure 4 in the revised manuscript because we felt that its information was redundant with Table 2 and took away from the significance of some of the other figures. We have updated our reported uncertainties throughout the manuscript and only identified significance at $p < 0.01$.

**Comment:** Figure 8. The color schemes should ideally be different between A and B, because they are not showing the same data groupings (e.g., the teal color in A doesn't represent the same data as the teal in B).
**Response:** Thank you for this suggestion. In the revised manuscript, we have significantly improved the presentation of Figure 8. We have reduced some redundancy and altered the color scheme between the $NO_x$ emission modeling by fuel and sector. Additionally, we split Figure 8 into two Figures (Figures 5 & 6) in the revised manuscript. This was done to separate the $NO_x$ emission modeling (Figure 5) from the $\delta(^{15}N)$ model (now Figure 6).

**Comment:** For Figures 3+4 and 6+7, the same data is shown but in different forms. However, the grouping is different between the paired plots, as Fig 3 and 6 have columns by site and Fig 4 and 7 have columns by $NO_3$ type. If it does not substantially affect the visual point you are trying to make with the plots, it would make it easier for the reader if the layout was the same for both paired plots.
**Response:** Thank you for this suggestion. In the revised manuscript, we removed Figures 4 and 7 from the text because we felt that the information was redundant with Tables 2, Figures 3 and 6, and the description in 3.1 & 3.2.